# Specialized interfaces of Smc5/6 control hinge stability and DNA association

Aaron Alt[1], Hung Q. Dang[2], Owen S. Wells[2], Luis M. Polo[1], Matt A. Smith[2], Grant A. McGregor[2], Thomas Welte[3], Alan R. Lehmann[2], Laurence H. Pearl[1], Johanne M. Murray[2] & Antony W. Oliver[1]

The Structural Maintenance of Chromosomes (SMC) complexes: cohesin, condensin and Smc5/6 are involved in the organization of higher-order chromosome structure—which is essential for accurate chromosome duplication and segregation. Each complex is scaffolded by a specific SMC protein dimer (heterodimer in eukaryotes) held together via their hinge domains. Here we show that the Smc5/6-hinge, like those of cohesin and condensin, also forms a toroidal structure but with distinctive subunit interfaces absent from the other SMC complexes; an unusual 'molecular latch' and a functional 'hub'. Defined mutations in these interfaces cause severe phenotypic effects with sensitivity to DNA-damaging agents in fission yeast and reduced viability in human cells. We show that the Smc5/6-hinge complex binds preferentially to ssDNA and that this interaction is affected by both 'latch' and 'hub' mutations, suggesting a key role for these unique features in controlling DNA association by the Smc5/6 complex.

[1] Cancer Research UK DNA Repair Enzymes Group, Genome Damage and Stability Centre, School of Life Sciences, University of Sussex, Falmer, BN1 9RQ, UK. [2] Genome Damage and Stability Centre, School of Life Sciences, University of Sussex, Falmer, Brighton BN1 9RQ, UK. [3] Dynamic Biosensors GmbH, Lochhamer Strasse, D-81252 Martinsried/Planegg, Germany. Correspondence and requests for materials should be addressed to L.H.P. (email: laurence.pearl@sussex.ac.uk) or to J.M.M. (email: j.m.murray@sussex.ac.uk) or to A.W.O. (email: antony.oliver@sussex.ac.uk).

Structural Maintenance of Chromosomes (SMC) complexes maintain genome integrity by regulating the organization, duplication and segregation of chromosomes in all kingdoms of life. In eukaryotes, cohesin maintains proximity and alignment of sister chromatids during and after S-phase, while condensin contributes to the formation of distinct compacted chromatids during prometaphase and metaphase. The third eukaryotic complex, Smc5/6, is essential in yeasts[1] and embryonic lethal when deleted in mice[2].

The role of Smc5/6 in the cellular response to DNA damage has been studied extensively. It is required for the resolution of recombination intermediates formed during mitosis[3–7] and meiosis[8–12], and for accurate chromosome segregation after replication stress[13].

While bacterial SMC proteins are generally homodimers, eukaryotic SMC complexes are based around specific heterodimeric pairs—Smc1/Smc3 (cohesin), Smc2/Smc4 (condensin) and Smc5/Smc6. All SMC proteins, whether homo- or heterodimeric, share a common architecture. Globular domains from the N and C-termini that respectively, provide the A and B motifs of a Walker ATPase, associate to form the 'head domain'. The two halves of the head are connected by a long anti-parallel coiled-coil 'arm' approximately 50 nm in length, capped by a 'hinge' domain where the coiled-coils reverse direction (Fig. 1a). The head domains of SMC dimers are bridged by a 'kleisin' component[14], such that full SMC complexes appear as closed ring structures in electron microscopy (EM)[15]. This has led to the proposal that SMC complexes function by encircling one or more DNA duplexes[16,17]. Association of SMC head domains generates two separate pockets, which, upon ATP-binding and hydrolysis, can dynamically regulate the opening and closing of the SMC complex 'ring'[18,19]. A number of observations suggest that the hinge interface is also able to open, and may be the site through which DNA duplexes are initially loaded[16]. Alternative models suggest that DNA-binding at the hinge promotes conformational changes leading to DNA loading through the heads[20,21].

Smc5/6 is the most elaborate member of the SMC-family, with six non-SMC elements (Nse or NSMCE, in yeast and humans, respectively) required for its biological activity[22,23]. The kleisin Nse4 forms a subcomplex with Nse1 and Nse3, which also possesses E3 ubiquitin ligase activity via the ring-finger domain of Nse1 (ref. 24). Nse2, an E3-SUMO ligase, binds to the coiled-coil 'arm' of Smc5 (refs 23,25). Both E3 ligase activities are required for some, but not all, biological functions of Smc5/6 (refs 26–28). Two further components, the HEAT-repeat proteins Nse5 and Nse6, are essential in budding yeast (but not in fission yeast) with orthologues recently identified in humans[27–29].

We have determined the X-ray crystal structure of the heterodimeric hinge of Schizosaccharomyces pombe Smc5/6 at a resolution of 2.75 Å. Despite low amino acid sequence similarity, the Smc5/6-hinge adopts the same toroidal architecture as other SMC-hinges, but possesses a distinctive 'molecular latch' feature at one of its two interfaces, which is conserved in Smc5/6 but absent in the other SMC systems. Mutagenic disruption of the latch and a second 'hub' feature severely impairs Smc5/6 function in vivo, and directly affects DNA interaction in vitro, suggesting that both features play important roles in DNA-capture and conformational switching of the Smc5/6 complex.

## Results

**Structure of the Smc5/6-hinge.** We obtained crystals of the Smc5/6-hinge from recombinant S. pombe proteins co-expressed in E. coli. The structure of the selenomethionine-labelled complex was determined by the single wavelength anomalous dispersion method (Supplementary Fig. 1a,b).

The structural information obtained led to the rational design of additional expression constructs. All biochemical and biophysical experiments used 'extended-hinge' constructs except for analytical size exclusion chromatography (aSEC), which used 'truncated-hinge' constructs (Supplementary Table 1 and Supplementary Fig. 1c). For clarity, figures have been simplified throughout, and display just the core fold of the Smc5/6-hinge (defined here as Smc5: amino acids 434–634, Smc6: 524–711) unless otherwise indicated.

The hinge of S. pombe Smc5/6 (Fig. 1b) has a similar structural architecture to that previously described for murine condensin (Smc2/4)[30] and murine cohesin (Smc1/3)[30,31] and the homo-dimeric SMC orthologues from Thermotoga maritima and Pyrococcus furiosus[18,32]. It forms a toroidal structure, in which the individual Smc5 and Smc6 hinge-domains interact through two distinct interfaces, termed North and South[33]. As in other SMCs, each hinge-domain is itself comprised of two subdomains—related by pseudo-twofold symmetry[18]—

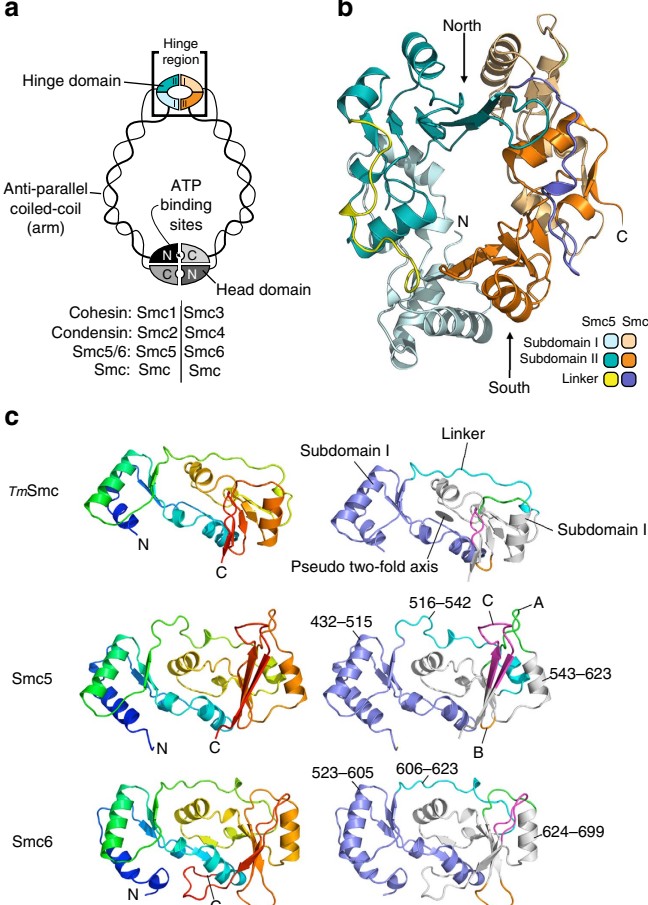

**Figure 1 | The heterodimeric hinge of S. pombe Smc5/6.** (a) Schematic diagram highlighting the conserved architecture and domain composition of the SMC family of proteins. (b) Molecular-cartoon depiction of the S. pombe Smc5/6 heterodimeric hinge, indicating component subdomains, and North and South interfaces; see associated key for details. (c) Comparison of the hinge-domains of Smc5 and Smc6 with a prototypical SMC protein from T. maritima (PDB: 1GXL). (Left) Molecular-cartoon depictions coloured blue → red, from N → C-terminus. (Right) Cartoons coloured according to subdomain, connecting loops and linker regions; see associated key for details. Amino acid boundaries for Subdomain I, Subdomain II and inter-connecting linker region are indicated.

connected together by an extended 'linker' (Fig. 1c). Structural comparison with other SMC-hinges reveals substantial divergence of both Smc5 and Smc6, particularly in the loops that connect the last three β-strands of Subdomain II—Loop A (Smc5: 585–597, Smc6: 664–671), Loop B (Smc5: 600–603, Smc6: 676–683) and Loop C (Smc5: 607–619, Smc6: 686–695). In contrast, the loops of Subdomain I are more conserved in length and closely resemble those of the other SMC proteins (Supplementary Fig. 2).

**A conserved loop stabilizes a divergent hinge interface.** Loop C of Smc5 (amino acids 607–619) constitutes a structural feature not seen in other SMC proteins (Fig. 2a). It appears to function as a 'molecular latch', forming an extended β-hairpin that makes several interactions across the North interface, which include amino acids from both the linker and Subdomain II regions of Smc6 (Fig. 2a,b). The hydroxyl group of Smc5-Ser610 makes a hydrogen bond to the side chain of Smc6-Asn642, plus an additional contact via a bridging water molecule to Smc6-Asp609. Smc5-Tyr612 makes hydrogen-bond interactions with Smc6-Glu647 and Smc6-Lys648, and sits in a small hydrophobic depression, created by the surrounding Smc6 amino acids Phe611, Tyr613 and Ile641.

Multiple amino acid sequence alignments across the Loop C region show that Ser610 and Tyr612 are conserved, in identity and sequential arrangement, across yeasts, plants and metazoans (Fig. 2c). The residues forming the hydrophobic depression in Smc6—that receives Smc5-Tyr612 (amino acids 611–613, 'FDY')—are not as strongly conserved, but a short hydrophobic-polar-hydrophobic motif is still maintained (Fig. 2d). This level of conservation suggested an important role for Loop C in Smc5/6 function, which we tested by introducing defined mutations into the *smc5* gene of *S. pombe* (by cassette-exchange at the endogenous locus[34]) and examining the resultant integrants for sensitivity to a range of DNA-damaging agents. We confirmed expression of the exchanged gene by western blot (see Supplementary Fig. 3a for this, and the other mutants described in this study).

When compared to wild-type yeast, strains containing the mutation of Smc5-Tyr612 to glycine (Y612G) displayed a mild temperature sensitivity when grown at 36 °C (Supplementary Fig. 3b) and at all concentrations and doses tested, were highly sensitive to camptothecin (CPT), hydroxyurea (HU), methyl methanesulfonate (MMS) and UV-irradiation (Fig. 3a). Mutation of Smc5-Ser610 to glycine (S610G) produced a weaker phenotype, with some sensitivity to all agents tested. However, the S610G/Y612G mutant was lethal, as no viable haploid strains could be obtained from sporulation of the *smc5*[+]/*smc5*-S610G Y612G heterozygous diploid.

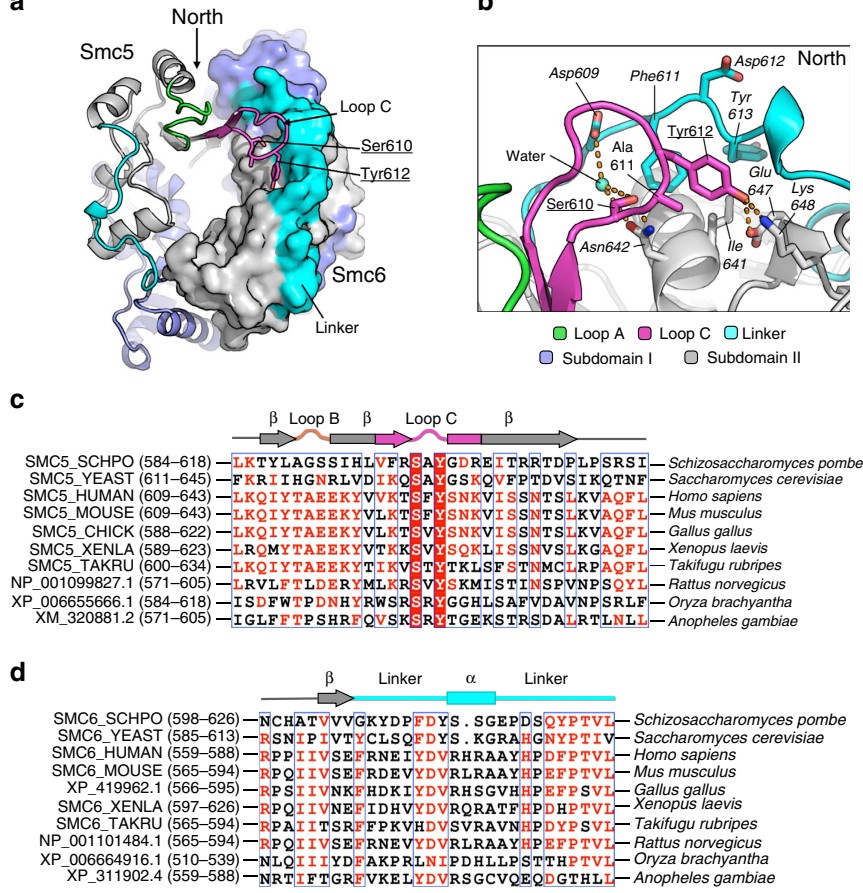

**Figure 2 | Smc5-Loop C is a conserved feature.** (**a**) Combined molecular cartoon (Smc5) and molecular surface (Smc6) highlighting the position of Smc5-Loop C and key amino acids. (**b**) Representative view of the molecular interactions made by Smc5-Loop C with the linker and Subdomain II regions of Smc6; see associated key for details. Selected amino acid residues are labelled. In this, and all subsequent figures residues from Smc5 are shown in plain type, and those from Smc6 in italic type. Amino acids mutated in this study are additionally underlined. (**c**) Multiple amino acid sequence alignment generated with MultAlin (http://multalin.toulouse.inra.fr/multalin/) for the Loop C region of Smc5. Highly conserved residues are indicated by a red background and white text. (**d**) MultAlin alignment for the linker region of Smc6. Amino acid residues conserved in physiological properties are coloured in red. Regions of conservation are indicated by the blue outline.

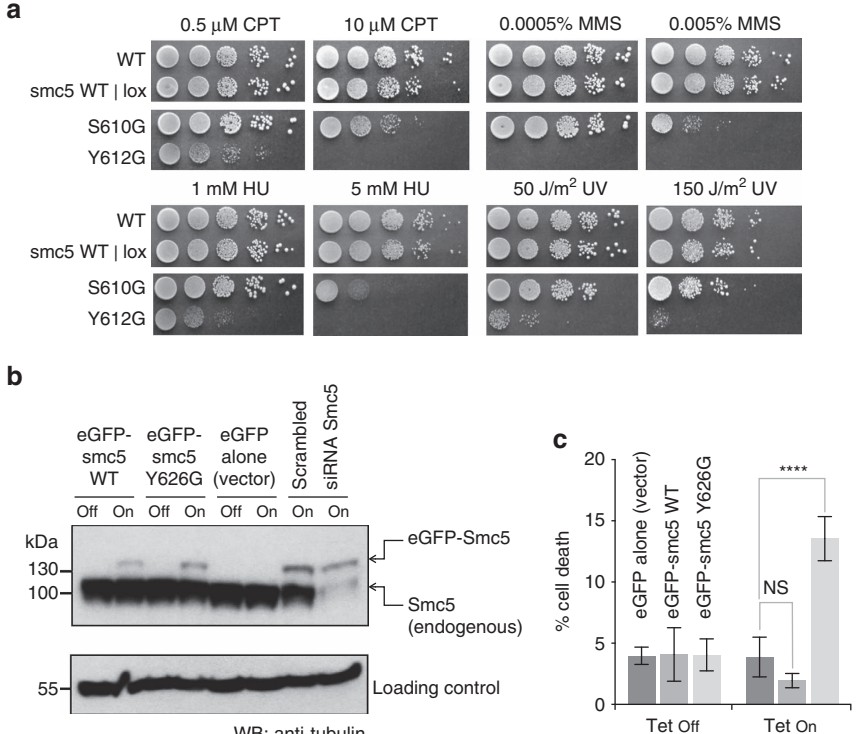

**Figure 3 | Mutation of Smc5-Loop C in yeast and human cells.** (**a**) DNA damage sensitivity of *S. pombe* strains containing Smc5-Loop C mutations S610G and Y612G. Dose and type of treatment is as indicated. WT 501 and Smc5 WT | lox strains are included as controls. (**b**) Anti-Smc5 western blot confirming induction of eGFP-fused proteins upon addition of doxycycline to the cell culture medium. Identities of species detected by the antibody were confirmed by treatment of cells with siRNA targeting Smc5, reducing the total amount of endogenous WT protein, but not affecting levels of the siRNA-resistant eGFP-fused Smc5. (**c**) Cell viability assay for U2OS cells stably transfected with doxycycline-inducible constructs expressing eGFP-fused wild-type or Loop C mutant (Y626G) forms of human Smc5. Results are the mean of three independent experiments, each in triplicate, with error bars representing 1 s.d. **** $P < 0.0001$, two-way ANOVA.

To determine whether Smc5-Loop C was also critical for function of human Smc5/6, we stably transfected U2OS cells with doxycycline-inducible constructs, which expressed either eGFP-fused wild-type or mutant human Smc5 (Fig. 3b). In the absence of doxycycline, all cells grew normally. When doxycycline was added, cells over-expressing either the empty vector control or the wild-type protein grew normally, but those expressing the Loop C mutant (Smc5-Y626G; the human equivalent to *S. pombe* Y612G) produced a highly significant increase in cell death (Fig. 3c).

Together, these data indicate that the 'molecular latch' formed by Smc5-Loop C is essential for cellular growth in both human and yeast cells, and that it represents a conserved and distinctive feature of the Smc5/6 complex.

**Conserved glycine motifs**. We note that a conserved arrangement of glycine residues is normally found in the hinges of the SMC family[35,36]; in eukaryotes, they fit a GX$_6$GX$_3$GG sequence motif, and in prokaryotes GX$_5$GGX$_3$GG (Fig. 4a,b). In each case, the glycines lie within the last two β-strands of Subdomain II and form an integral part of the dimerization interface, such that dimerization of the hinge domains can be disrupted *in vitro* by mutation of the conserved glycines[35].

In Smc5/6, the North interface (Fig. 4c, left) lacks glycines at any of the expected positions, while the South interface contains a partial match—with the first two glycine residues found at the expected positions—albeit not strictly within the previously defined consensus motif. The first glycine (Gly683) is instead separated from the second (Gly694) by a gap of 10 amino acids, rather than the eukaryotic consensus of six, due to the insertion of amino acids within loop C of Smc6-Subdomain II (Fig. 4b,c, right).

**Hinge-domain association**. We hypothesized that mutation of Smc5-Loop C would specifically disrupt the North interface. To provide a suitable control, we also generated a series of mutations along the last β-strand of Smc6 (centred around the conserved glycine residue Gly694), designed to sterically disrupt the South interface (6-Mut: Smc6-S692E, -G694K, -S696E; see Supplementary Fig. 4 for additional details). Cell lysates from *E. coli* co-expressing recombinant protein were passed through an immobilized metal affinity chromatography (IMAC) column, to retain the Smc5 hinge-domain (His$_6$ affinity-tag), and then probed by western blot for co-purification of the Smc6 hinge-domain (Strep$_{II}$ affinity-tag) (Fig. 4d). While we saw robust co-purification of wild-type hinge-domains (lane 13)––as predicted, association of the hinge-domains was compromised by the introduction of either Loop C or 6-Mut mutations (lanes 14, and 15, respectively), and was completely abolished when both sets of mutations were combined (lanes 16).

***smc6-X* and *smc6-T2* map to a single nexus**. Our structural information enables us to examine in detail two single point mutations, known to occur within the hinge of Smc5/6, which both have impact on cellular function: *smc6-X*[37], the first identified Smc6 mutant, in which Smc6-Arg706 is mutated to cysteine[38], and the temperature sensitive (*ts*) mutant *smc6-T2* in which Gly551 is mutated to arginine[23]. Arg706 is positioned at

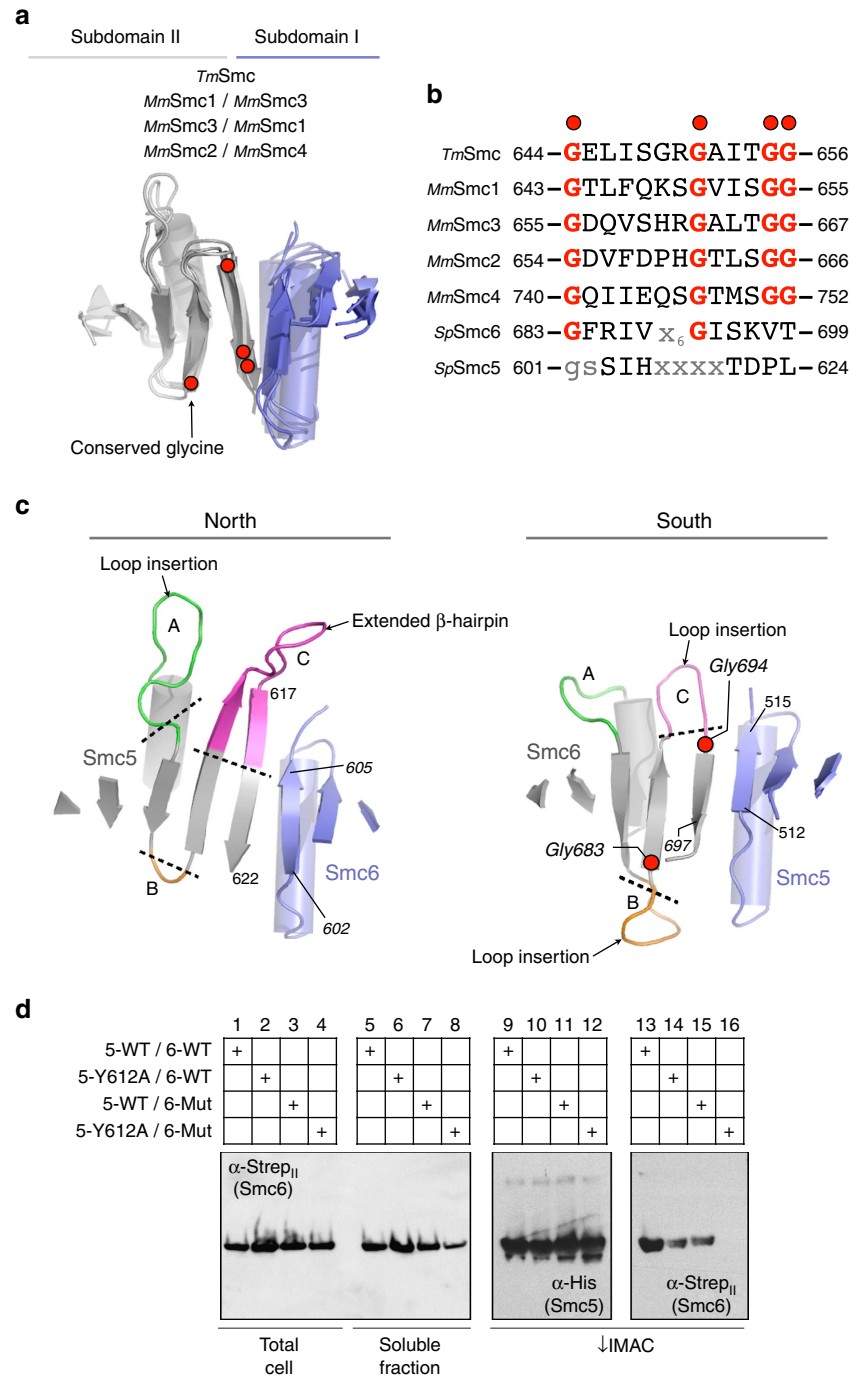

**Figure 4 | The heterodimer interfaces of Smc5/6 are highly divergent. (a)** Schematic secondary structure molecular cartoon, highlighting both the 8-stranded β-sheet and the position of conserved glycine residues, found at the core of each hinge interface in murine cohesin (Smc1/3), murine condensin (Smc2/4) and *Tm*Smc. **(b)** Amino acid sequence alignment highlighting the conserved set of glycine residues found in the last two β-strands of Subdomain II of SMC-family hinge-domains. Smc6 contains a partial match to the consensus sequence, but Smc5 does not. **(c)** Molecular cartoon, as in **a** but for the North and South interfaces of Smc5/6. The loops connecting the last three β-strands of Subdomain II are additionally highlighted, and labelled consecutively from A to C. Amino acids at the start and end of the β-strands that pair to form each interface are numbered. **(d)** Assessing hinge stability by co-expression/co-purification assay. His-tagged Smc5-hinge was co-expressed with Strep<sub>II</sub>-tagged Smc6-hinge in *E. coli*. After lysis, and clarification, the soluble fraction was passed through an IMAC column, capturing the Smc5-hinge. After successive washes, to remove any unbound material, the amount of co-purified Smc6-hinge was assessed by western blot. WT = wild-type, 5-Y612A = Smc5-hinge containing the Loop C mutation, 6-Mut = Smc6-hinge containing S692E, G694K and S696E mutations.

the C-terminal end of the Smc6 hinge-domain, just before the start of the exiting coiled-coil. The guanidinium head group of this residue forms part of an extensive hydrogen bond network, making contacts with the backbone carbonyls of Glu569 and Gly573, as well as the side chains of Glu569 and Asn577 (Fig. 5a, left). It also forms part of an unusual cluster of arginine residues (spatially arranged around Trp701), which also includes Arg570 and Arg703.

Smc6-Gly551 is located at the end of the first β-strand of Subdomain I, and is immediately followed by Pro552 (Fig. 5a).

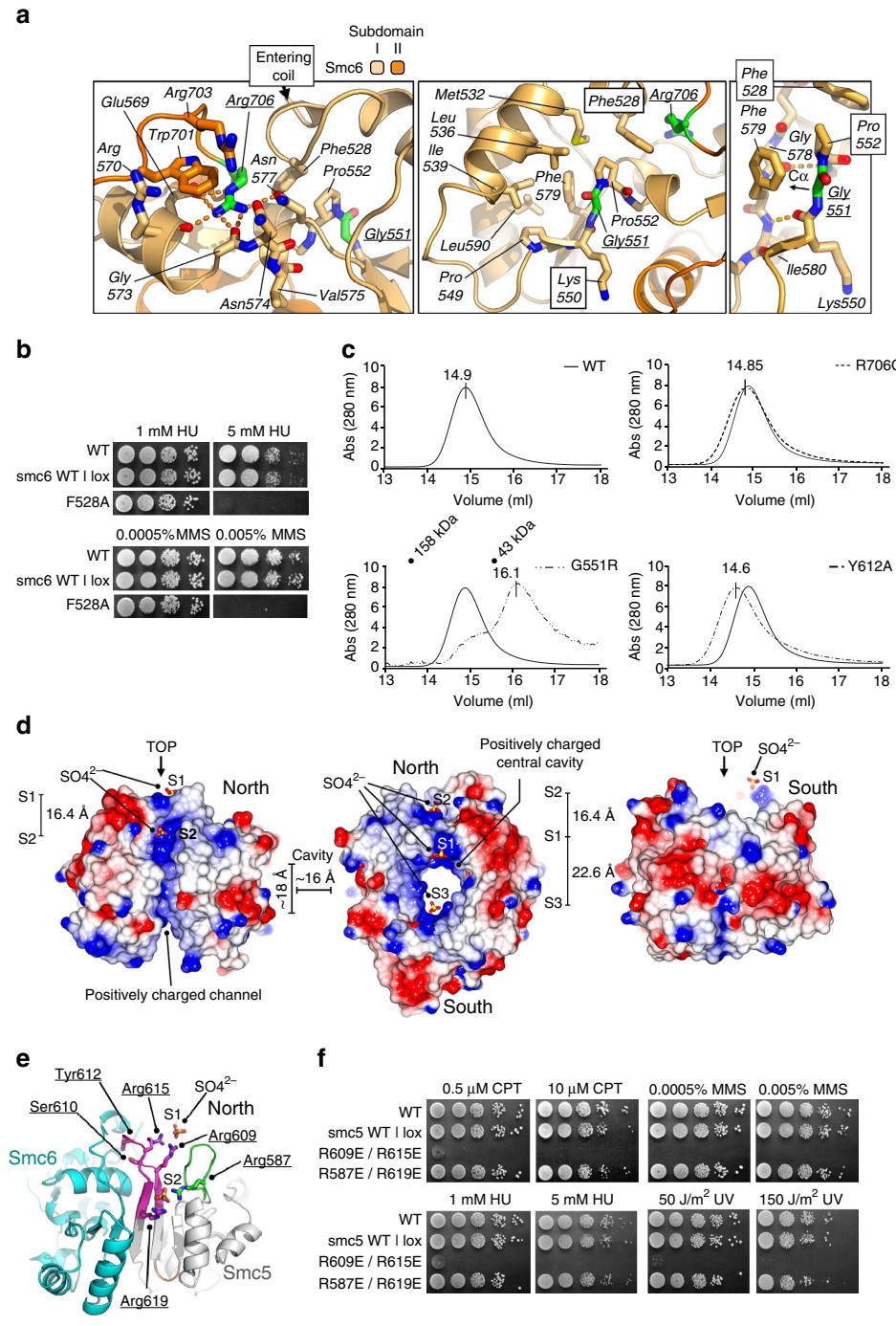

**Figure 5 | Alleles of smc6 and sulphate-ion coordinating arginine-pairs** (**a**, left) Molecular secondary structure cartoon, highlighting the relative positions of the Arg706 and Gly551 residues (coloured green) mutated in the *S. pombe smc6-X* and *smc6-T2* alleles, respectively. (**a**, middle) Alternative view of the Gly551 residue, showing its proximity to Arg706. (**a**, right) Mutation of Gly551 to any residue would cause disruption of the protein fold, due to the Cα position being conformationally constrained to point into the core of the protein, rather than out to solvent. Potential hydrogen bonds are indicated by the orange dashed-lines. See associated key for additional details. (**b**) DNA damage sensitivity of *S. pombe* strains containing the single-point mutant F528A. Dose and type of treatment is as indicated. (**b**) Spot tests showing HU and MMS sensitivity of Smc6-F528A strain. (**c**) Analytical size exclusion chromatograms for wild-type or the indicated mutant forms of Smc5/6-hinge. 'Truncated-hinge' constructs were used in these experiments (see Supplementary Table 1) as we found that elution volume differences were more pronounced than in equivalent experiments with extended hinges; this is consistent with a dominant effect of the long flexible 'arms' on hydrodynamic radius, masking subtler changes in hinge conformation. The elution peak positions of a molecular mass calibration are also shown for the G551R (*smc6-T2*) chromatogram. (**d**) Molecular surface representations, coloured by electrostatic potential. (Left) view of the North interface, (Middle) top down view, (Right) view of the South interface. Bound sulphate ions are shown in ball-and-stick representation, and consecutively labelled S1 – S3. (**e**) Molecular secondary structure cartoon highlighting the position of the Smc5 Arg587/Arg619 and Arg609/Arg615 pairs which each coordinate a sulphate ion. See associated key for details. (**f**) DNA damage sensitivity of *S. pombe* strains containing the charge-reversal mutants R587E/R619E and R609E/R615E. Dose and type of treatment is as indicated.

Both amino acids form an integral part of a hydrophobic cluster at the core of this part of the protein—comprised of Smc6 residues Met532, Leu536, Ile539, Pro549, Phe579 and Leu590. Pro552 also provides the cap for the start of the following α-helical element. Interestingly, the side chain of Arg706 is packed up against that of Phe528, which is itself packed against Pro552, which follows Gly551.

We predicted that mutation of Phe528 in Smc6 would also have functional consequences in yeast, as it also forms an integral part of the Arg706/Phe528/Pro552/Gly551 nexus or 'Smc6-hub' (Fig. 5a). Specifically, Phe528 appears to act as a single anchor point for the incoming coiled-coil, which forms part of the Smc6 'arm', and connects it to the rest of the hinge-domain. The F528A mutation, when introduced into yeast, conferred sensitivity to both HU and MMS treatment, but not to UV, CPT, or to elevated growth temperature (Fig. 5b and Supplementary Fig. 5a). Unlike the latch-mutation (Y612A) these phenotypes were not produced by disruption of hinge-dimerization, as confirmed by co-purification experiments (Supplementary Fig. 5b). Mutation of the equivalent residue in Smc5 (W436A) had no observable phenotype (Supplementary Fig. 5a) under any of the conditions tested; this is compatible with the more complex, predominantly hydrophobic interface found in Smc5, disruption of which would require several mutations (Supplementary Fig. 5c).

Together these data indicate the importance of the 'Smc6-hub', and allude to the presence of functional asymmetry within the Smc5/6 complex.

**Dimerization analysis by aSEC.** We next sought to confirm the effects of selected mutations on hinge-dimerization by aSEC. The wild-type hinge eluted as a single heterodimeric peak from an aSEC column at a volume of 14.9 ml (Fig. 5c). Complex containing Smc6-R706C also eluted as a single peak at a similar volume of 14.85 ml, confirming a previous observation[38] that this mutation does not grossly affect protein fold or prevent hinge-dimerization. In contrast, the elution profile of the hinge containing Smc6-G551R contained two distinct peaks, with the major species eluting at a larger volume of 16.1 ml. From molecular mass calibrations, this corresponds to the expected elution point of individual monomers. Our data are therefore consistent with: G551R causing a gross disruption of protein fold; the observed *in vivo* temperature sensitivity of yeast strains containing this mutation; and previous *in vitro* transcription-translation/immunoprecipitation experiments, in which the mutation abolished the interaction between epitope-tagged Smc5 and Smc6 hinge-domain constructs[23].

We also tested hinge-dimerization of the Smc5-Y612A Loop C latch mutant. This elutes as a single peak from aSEC, but at a slightly lower elution volume (14.5 ml) than the wild-type complex, consistent with the hinge displaying a larger apparent hydrodynamic radius, due to opening at the North interface, but with the South interface still intact.

**Bound sulphate ions and potential nucleic-acid binding sites.** We noted that, in Smc5/6-hinge crystals, a number of sulphate ions (supplied from the crystallization mother liquor) were bound to both molecules of the asymmetric unit (Fig. 5d). As sulphate is *iso*-structural with phosphate, we speculated that the position of each ion could indicate a potential path for the phosphodiester backbone of a bound DNA molecule, especially as several laboratories have shown that the hinges of other members of the SMC family are capable of binding to nucleic acid[30–32,36,39].

Like other members of the SMC family[31] the central cavity of the Smc5/6-hinge is positively charged (Fig. 5d, centre).

However, uniquely to Smc5/6, the central cavity is directly connected to a positively charged channel or groove that runs along the North interface (Fig. 5d, left). In contrast, the South interface is predominantly neutral (hydrophobic)/negative in charge (Fig. 5d, right).

Two pairs of Smc5 residues caught our attention, each of which coordinated a single sulphate ion (S1 and S2; Fig. 5e): Arg587/Arg619 flank the positively charged channel, whereas Arg609/Arg615 are located on Smc5-Loop C in close proximity to both Ser610 and Tyr612 of the latch (Fig. 5e). We mutated both arginine pairs to glutamic acid (reversing the formal charge at these positions) and again assayed for cell viability and DNA-damage sensitivity (Fig. 5f, Supplementary Fig. 3b). The R609E/R615E mutant was temperature sensitive with a slow-growth phenotype at 25 and 30 °C, and high sensitivity to CPT, MMS, HU and UV, when compared to wild-type controls. Again, we could confirm that these phenotypes were not produced by disruption of hinge-dimerization, or by gross reductions in expression level (Supplementary Figs 3a and 5b). In contrast, the R587E/R619E mutants were viable at all temperatures and were insensitive to the agents tested.

**The Smc5/6-hinge binds preferentially to ssDNA.** We then investigated association of the Smc5/6-hinge with DNA using fluorescence polarization. We initially chose to examine an oligonucleotide length of 45 nt, as this was reported to be the minimal length required for stoichiometric interactions of recombinant full-length *Saccharomyces cerevisiae* Smc5 and Smc6 proteins with ssDNA[40,41].

The hinge bound preferentially to a 45 nt single-stranded oligonucleotide, with a $K_d$ of $\sim 2 \mu M$ (Fig. 6a). Binding to a 45 bp DNA duplex also occurred, but with far lower affinity (even at 10 times the $K_d$ for ssDNA, that is, 20 μM, only $\sim 45\%$ of dsDNA was bound). Interaction of the hinge with ssDNA was also evident in an electrophoretic mobility shift assay (EMSA), where a single fluorescent-labelled retarded species could be seen at moderate protein concentrations (0.9 and 1.8 μM; labelled 'Complex' in Fig. 6b). However, at higher protein concentrations ($> 1.8 \mu M$) there was a second species (labelled with an asterisk, Fig. 6b) that remained in the well even after prolonged electrophoresis. The material in the well could also be seen by the naked eye—indicative of a highly aggregated protein-DNA species (Fig. 6b inset). Despite this, it was still possible to calculate a $K_d$ for the interaction of the Smc5/6-hinge with the 45 nt oligonucleotide ($\sim 1.2 \pm 0.7 \mu M$) and directly compare this to the value obtained by fluorescence polarization ($\sim 2 \pm 0.03 \mu M$; Supplementary Fig. 6a). The two values are in reasonable agreement, when errors in experimental fit are considered.

We went on to examine the minimum length of ssDNA sufficient for Smc5/6-hinge interaction. We used 5′-biotinylated oligonucleotides of different lengths, bound to magnetic beads, as 'bait' in pull-down experiments using WT Smc5/6-hinge as 'prey' (Fig. 6c). We selected the 15mer as the preferred substrate for subsequent fluorescence polarization experiments, as it retained a similar amount of hinge as the longer lengths of ssDNA tested in the pull-down experiment (as judged by intensity of staining, Fig. 6c). WT Smc5/6-hinge bound the 15mer with a calculated dissociation constant ($K_d$) of $\sim 1.2 \mu M$. Smc5-Y612A ($\sim 1.3 \mu M$), Smc6-F528A ($\sim 1.7 \mu M$) and Smc6-R706C ($\sim 1.6 \mu M$) each bound with similar, or moderately lower affinity, whereas Smc5-R609E/R615E significantly disrupted DNA-binding ($K_d$ = ND, not determined, Fig. 6d).

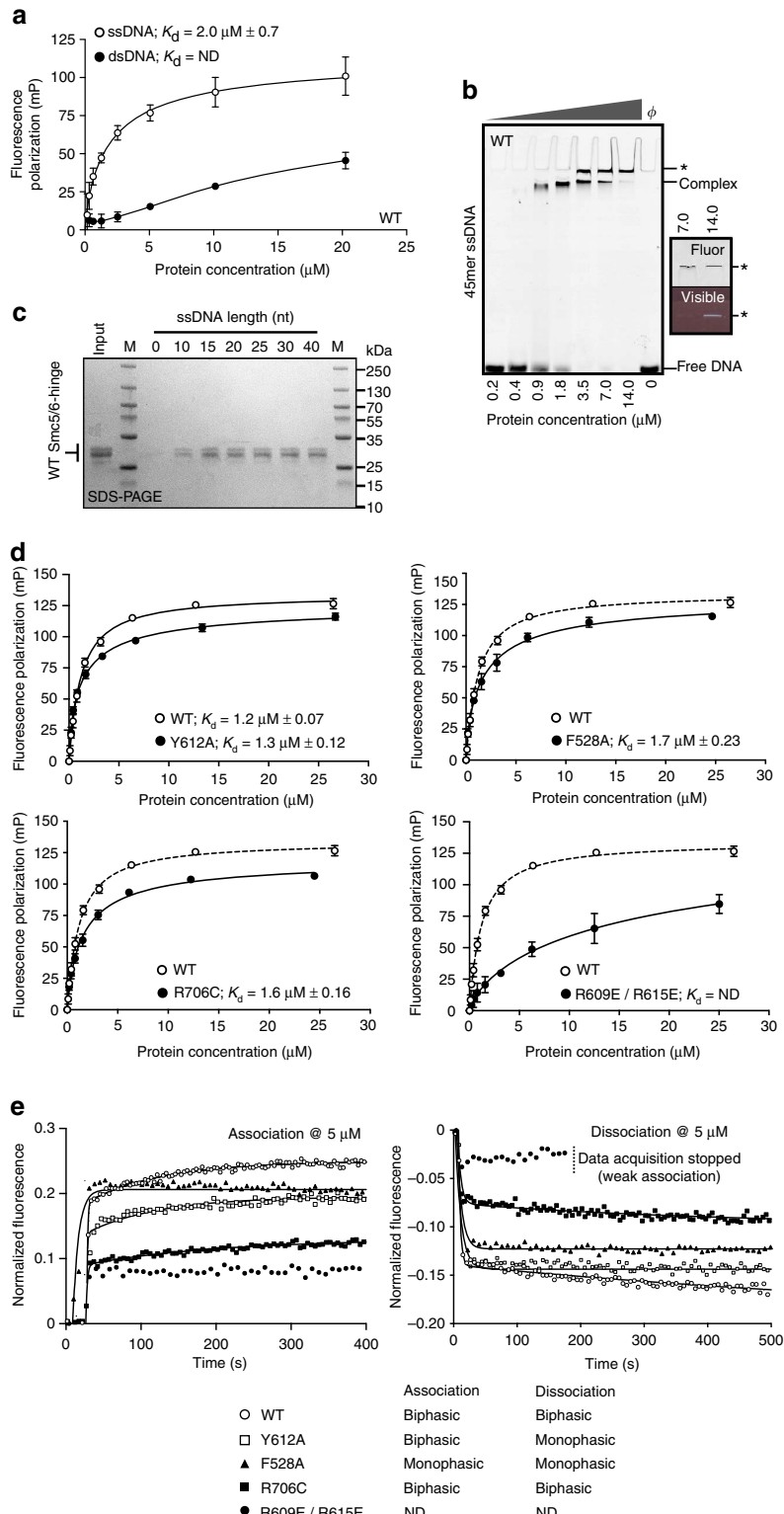

**Figure 6 | The Smc5/6-hinge preferentially binds to ssDNA.** (**a**) Fluorescence polarization assay. Wild-type (WT) Smc5/6-hinge binds with higher affinity to a 45 nt ssDNA (open circle) substrate than to a 45 bp dsDNA substrate (closed circle). (**b**) Representative EMSA gel, imaged by laser-scanning, for the interaction of wild-type Smc5/6-hinge with a 45 nt ssDNA substrate. The position of a fully resolved protein:DNA species is indicated by the 'Complex' label. At higher protein concentrations a second species, which does not enter the gel, is also observed (labelled with an asterisk). (**c**) Biotinylated-oligonucleotide pull-down assay. WT Smc5/6-hinge was incubated with magnetic beads coated with the indicated length of oligonucleotide. M = molecular mass marker. (**d**) Fluorescence polarization assays for binding of the indicated Smc5/6-hinge mutant to a 15mer oligonucleotide. (**e**) Sensorgrams for SwitchSENSE 'F$_{down}$' experiments. Representative association (left) and dissociation curves (right), at a protein concentration of 5 μM, are shown for binding of WT and mutant forms of the Smc5/6-hinge (top) to immobilized ssDNA. The solid black lines indicate the fit of the indicated binding model to the experimental data. Fluorescence polarization data are the mean of either 4 (**a**) or 3 replicates (**d**,**e**) with error bars indicating 1 standard deviation. Dissociation constants (K$_d$) were calculated by least-squares fitting of a one-site binding model. ND, not determined.

**Kinetics of ssDNA-binding.** To directly examine kinetics of DNA-binding, we used SwitchSENSE technology (Dynamic Biosensors GmbH, Martinsried, Germany; see Methods). A single biochip, to which a single-stranded 48mer was immobilized, was used for all experiments. (NOTE: shorter lengths of DNA are not compatible with the experimental setup).

WT Smc5/6-hinge readily associated with, and dissociated from, the immobilized ssDNA (Fig. 6e and Supplementary Fig. 6b). As in fluorescence polarization, DNA-binding was significantly disrupted by the Smc5-R609E/R615E mutant. In contrast, however, each of the remaining mutants also perturbed ssDNA-binding to some extent, as visualized by the reduction in the maximum level of fluorescent signal achieved; exemplified by the Smc6-R706C mutant.

It was not possible to use a simple 1:1 Langmuir binding model to consistently fit all the experimental data. For example, a more complex biphasic association/biphasic dissociation model was required for the WT protein. This indicates that the mode of ssDNA-binding by the Smc5/6-hinge is complex, and could involve more than one DNA interaction site, and/or some concomitant change in conformation.

As determining accurate kinetic parameters for biphasic data can be problematic, we chose to use the data more empirically; examining the shapes of the binding curves and asking whether a monophasic or biphasic binding model provided an overall better fit to the experimental data. Both Smc5-Y612A and Smc6-F528A mutant forms of the hinge produced altered binding profiles. While Smc5-Y612A associated with DNA in a manner similar to the WT protein (biphasic), it dissociated with monophasic behaviour. Moreover, Smc6-F528A both bound and dissociated from DNA with monophasic kinetics, and also appeared to associate with ssDNA at lower protein concentrations than the WT Smc5/6-hinge (Fig. 6e, Supplementary Fig. 6b). In both cases, functional aspects of ssDNA-binding by the Smc5/6-hinge have been directly affected by the introduced mutations.

**An 'arms-closed' conformation in solution.** Crystal structures of 'extended-hinges' from *Pyrococcus furiosus* SMC (*Pf*SMC) and *Saccharomyces cerevisiae* Smc2/Smc4 (*Sc*Condensin)—and the structurally related protein MukB from *Escherichia coli* (*Ec*MukB)—indicate that several types of 'arm' architecture are possible in solution. *Ec*MukB adopts an 'open' conformation, with widely splayed arms[42], whereas *Pf*SMC and *Sc*Condensin are in 'closed' conformations with tightly associated arms[43] (panel *i*, Fig. 7a). We therefore measured the small-angle X-ray scattering (SAXS) profile of the Smc5/6-hinge in order to examine which architecture it adopts in solution (Supplementary Fig. 7).

Goodness-of-fit comparisons between calculated SAXS profiles and experimental data indicate a high degree of shape/conformational similarity between Smc5/6-hinge and *Sc*Condensin, $\chi = 1.43$ (chi value, FoXS[44]), but poor similarity to either *Pf*SMC or *E. coli* MukB ($\chi = 4.68$ and 8.80, respectively) (panel *ii*, Fig. 7a). The pair-distance distribution function [$P(r)$-distribution] calculated from the scattering data (panel *iii*, Fig. 7a) is also consistent with Smc5/6-hinge adopting an extended conformation in solution[45] ($D_{max} = \sim 156$ Å) and resembles that back-calculated from the *Sc*Condensin structure[46]. A single *ab initio* dummy atom model produced the best $\chi$ value of 0.47 (panel *ii*, Fig. 7a). The molecular 'envelope' described by this model strongly resembles the closed-arm conformation of *Sc*Condensin (panel *iv*, Fig. 7a); a folded, asymmetric, rod-like structure, with the hinge tilted to one side relative to the axis of the tightly paired coiled-coil arms (Fig. 7b, adapted from ref. 43).

Soh *et al.*[43] describe part of the incoming Smc4 coil as a 'rooting helix', due to the extensive hydrophobic interactions it makes with the main body of the Smc4 hinge that help to lock *Sc*Condensin into the closed, tilted hinge configuration (Fig. 7c and inset). Mutations in either the rooting helix (Leu676Glu) or in the receiving part of the hinge (Leu731Glu) were lethal in budding yeast, showing this 'locking' function to be essential[43]. While the arrangement of molecules forming the lattice of Smc5/6-hinge crystals is complex (see Supplementary Fig. 1), the similarity between the positions of the two incoming N-terminal helices of Smc5/6 and those of *Sc*Condensin is striking. Moreover, the same 'tilted-hinge' conformation observed for *Sc*Condesin is also evident in both molecules of the Smc5/6-hinge asymmetric unit (Fig. 7b,c). This, along with our SAXS analyses, strongly reinforce the suggestion that this is a biologically relevant conformation for the heterodimeric members of the SMC family.

Residues 502–540 of Smc6 encode the equivalent to the Smc4-rooting helix, but do not form a single α-helical element (Fig. 7c). Instead, a short helical segment (aa 524–529) serves to interrupt the incoming N-terminal coil. Significantly, amino acid Phe528 is located on this insert, making the series of hydrophobic interactions already described with the 'hub' of Smc6 (Arg706/Phe528/Pro552/Gly551). It is therefore likely that Smc6-Phe528 fulfils a similar functional role to that of Smc4-Leu676.

## Discussion

The X-ray crystal structure of the Smc5/6-hinge from *S. pombe* identifies two distinctive features: (a) the 'molecular latch' of Smc5, which stabilizes the 'North' interface, in lieu of the glycine-rich motif conserved at this location in other SMC-family members; and (b) the 'hub' of interacting residues in Smc6, which directly contacts the 'rooting' helix[43] and connects the incoming arm of Smc6 to the globular hinge domain. The presence of these distinctive features is consistent with the observed co-evolution of the Smc5 and Smc6 hinges away from the other SMC proteins and towards a more specialized function[47]. We also speculate here that the distinctive structure of the North interface indicates (and facilitates) a more dynamic loading/unloading requirement for the Smc5/6 complex compared to cohesin or condensin. Again, this is consistent with the involvement of Smc5/6 in comparatively transient processes such as stabilization of stalled replication forks and regulation of recombination[3,7].

Mutation of a highly conserved residue in the latch (Smc5-Tyr612, Tyr626 in humans), which mediates interaction with Smc6, disrupts stable heterodimerization of the hinge *in vitro*. Mutation of latch residues Smc5-Arg609 and Arg615, which interact with a well-ordered sulphate ion present in the crystallization medium, strongly disrupts association with all ssDNA substrates tested *in vitro*. All of these mutations generate loss-of-function phenotypes, suggesting that the molecular functions of heterodimerization and ssDNA-binding are functionally coupled *in vivo*.

The latch (Smc5-Loop C) delimits one end of a narrow channel that curves around the outer face of the North interface and connects it to the 'Smc6-hub' of residues centred around Phe528, and involves the residues affected by the previously described *smc6-X* (R706C) and *smc6-T2* (G551R) mutations[23,37]. As with the latch, mutations in the 'Smc6-hub' also produce loss-of-function phenotypes.

While the external channel is itself highly basic, and in the crystal structure contains an additional well-ordered sulphate ion (S2, see Fig. 5d), our current set of experiments cannot determine unambiguously if it is functionally relevant; charge reversal mutation of Smc5-Arg619 and Smc6-R587, residues that line the channel and coordinate the S2 ion, does not produce a phenotype

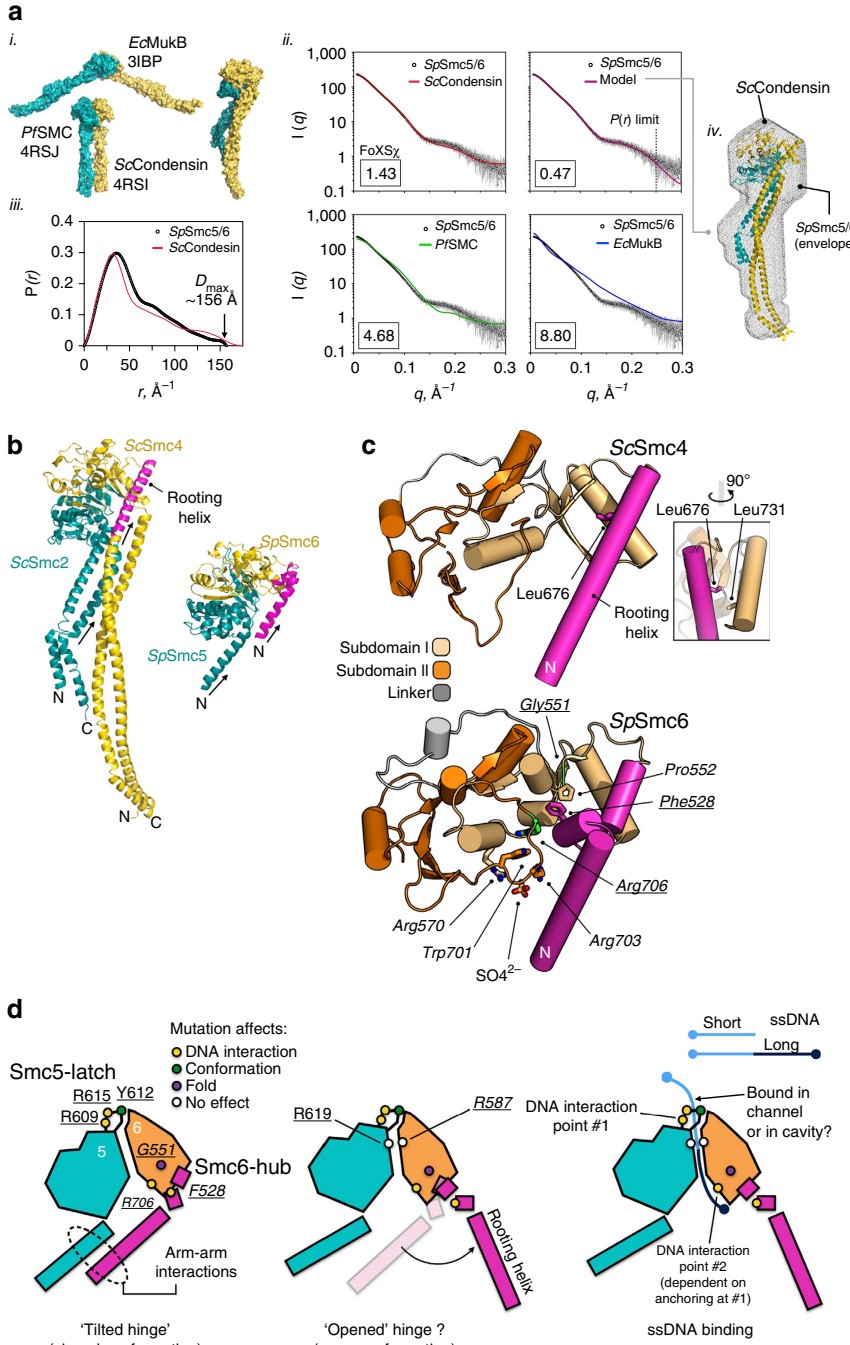

**Figure 7 | Conformation of the Smc5/6-hinge.** (**a**) (*i*) Molecular surface representations for 'extended-hinge' structures of *E. coli* MukB (*Ec*MukB), *P. furiosis* SMC (*Pf*SMC) and *S. cerevisiae* Smc2/Smc4 (*Sc*Condensin). One protomer of the hinge-dimer is coloured in cyan, the other in yellow. (*ii*) P(*r*)-distribution for the extended-hinge of Smc5/6 (open circles) and calculated P(*r*)-distribution for *Sc*Condensin (red line) (*iii*) Fits of calculated SAXS profiles for *Sc*Condensin, *Pf*SMC and *Ec*MukB, as well as the *ab initio* DAMMIF dummy atom model, to the experimental data collected for Smc5/6 (open circles). Goodness-of-fit values (chi, $\chi$) determined with FoXS[44] are shown in each case. (*iv*) Overlay of the extended-hinge structure of *Sc*Condensin with the molecular envelope (grey mesh) defined by the DAMMIF dummy atom model. (**b**) Molecular cartoon representations of the heterodimeric hinges from *Sc*Condensin[43] and Smc5/6-hinge (this manuscript), highlighting the positions of the incoming N-terminal coils (marked with arrows), and the 'rooting helix' (coloured magenta). (**c**) Schematic molecular cartoon of the hinge-domain of *Sc*Smc4 (top) and *Sp*Smc6 (bottom). The rooting helix of *Sc*Smc4 is anchored by the side chain of Leu676, which makes interactions with a hydrophobic cluster that includes Leu731 (inset). Mutation of either residue, to glutamic acid, is lethal in budding yeast. The rooting helix equivalent in *Sp*Smc6 is interrupted by a short helical element, which contains amino acid Phe528 that interacts with amino acids of the Smc6 'hub'. Positions of Arg706 and Gly551, mutated in the fission yeast alleles *smc6-X* and *smc6-T2*, respectively, are additionally highlighted (stick representation, with carbon atoms coloured green). See associated key for additional details. (**d**) A schematic and speculative model for DNA-mediated conformational changes at the Smc5/6-hinge, based upon the experimental work presented here, and that of Soh *et al.*[43]. The positions of key amino acids, forming the latch and hub features of Smc5/6 are highlighted, and colour-coded according to their observed effect on Smc5/6-hinge function or fold (see associated key).

*in vivo*, in the assays tested. However, if the function of the channel is simply to provide non-specific contacts with the phosphodiester backbone of a bound DNA molecule, it may be necessary to introduce a more extensive set of mutations in order to fully disrupt DNA-binding, that is, to alter the series of polar residues that line the channel, each of which could theoretically contribute to a larger composite DNA-binding surface.

We propose that the latch and hub features of the Smc5/6-hinge represent two distinct 'interaction points' for ssDNA-binding, with the charged channel (or internal cavity) providing only non-specific contacts with the phosphodiester backbone (Fig. 7d). Tight binding is clearly dependent on a primary interaction made with the arginine-pair of the latch (Smc5-R609/R615, DNA interaction point #1, Fig. 7d). When ssDNA of sufficient length is encountered and 'anchored' at the latch, a second interaction with residues of the Smc6-hub is then promoted (DNA interaction point #2). Any bound DNA could then theoretically gain entry to the central charged cavity in a gated-type mechanism[16,48,49].

Loss of binding at interaction point #1 leads to a severe growth defect in fission yeast, whereas loss of interaction point #2 (as defined by the well-characterized R706C smc6-X mutant) leads to sensitivity to replication stress and a range of DNA damaging agents[1,3,50]. This model is consistent with the biphasic association and dissociation kinetics observed for wild-type Smc5/6-hinge, when binding to longer lengths of ssDNA. This mode of binding is modified by mutations that affect either heterodimerization (Y612A, Smc5-latch) or anchoring of the 'rooting' helix (F528A, Smc6-hub). In particular, the latter mutation alters hinge behaviour such that it now binds to ssDNA with simple monophasic association and dissociation kinetics. It also appears to bind to ssDNA more rapidly, indicating that an 'energetic barrier' to DNA-binding has been removed (filled triangles, Fig. 6e).

Although the F528A mutation clearly affects ssDNA-binding and displays a strong *in vivo* phenotype, SAXS measurements indicate that it does not, of itself, alter the resting conformation of the hinge in solution; which depends on interactions additional to those made by the rooting helix and is likely to be strongly influenced by interaction with DNA[43] (Supplementary Fig. 8). An emerging body of evidence indicates that conformational changes at the hinge-to-arm junctions of SMC-family proteins is driven by, or arises as a result of, DNA-binding[39,43,51]. This event serves to communicate 'engagement' of the hinge to the rest of the complex, and promote (or be promoted by) ATP-hydrolysis at the distal head domains[51]. Such conformational changes could also be regulated by, or indeed serve to regulate, the ubiquitylation and SUMOylation activities of its Nse1 and Nse2 components, whose roles in the biological function of Smc5/6 are currently unclear.

The *in vitro* and *in vivo* effects of mutations in both latch and hub features of Smc5/6 suggest a complex set of interactions, with a concomitant set of conformational changes driven by DNA-binding. However, determination of a hinge-ssDNA structure will be required to confirm this.

## Methods

**Cloning.** DNA encoding the required region of *S. pombe* Smc5 or Smc6 was amplified by PCR, using synthetic DNA codon-optimized for expression in *E. coli* as a template (Genscript, Piscataway, USA). Primers were designed to include restriction sites, to facilitate sub-cloning of the amplified DNA into vectors suitable for protein expression in *E. coli*.

**Design and evolution of expression constructs.** As secondary structure predictions and multiple amino acid sequence alignments were mostly uninformative, sequence-threaded models—generated by the Phyre[2] web-server[52]—helped guide the creation of several expression constructs, exploring the domain boundaries of both *S. pombe* Smc5 and Smc6 hinge domains. By co-expression of each component in *E. coli*, we were able to identify those constructs that expressed soluble protein, formed a heterodimeric complex, and which could be purified by sequential chromatographic steps.

During the purification of one such complex, the presence of a co-purifying species was observed by SDS-PAGE. This was analysed by Edman degradation, and subsequently identified as a proteolytic product of Smc6. With this information, our expression constructs were re-designed, leading to Smc5-hinge (encoding amino acids 336–692) and Smc6-hinge (448–703) being used to express protein for initial crystallographic screening.

We obtained crystals of the purified complex and its structure was determined at 2.75 Å resolution, with phases provided from seleno-methionine labelled protein using the single wavelength anomalous dispersion method (SAD).

However, it became apparent, while refining the structure that our expression constructs actually encoded unmatched lengths of alpha-helix at both their N and C-termini—that is, corresponding to the helices that would normally be expected to pair forming an anti-parallel coiled-coil—which resulted in the C-terminal helix of Smc5 (residues 638–691) being erroneously paired with the N-terminal helix of Smc6 (residues 448–498) (Supplementary Fig. 1), but nevertheless forming crucial intermolecular contacts required to build the crystal lattice.

With this additional structural information, we were able to rationally redesign our expression constructs (Supplementary table 1).

**Expression constructs for crystallization.** DNA encoding amino acids 336–692 of Smc5 was cloned into an in-house modified version of pCDF-1b (Merck Millipore, Darmstadt, Germany) that expresses the recombinant protein with an N-terminal human rhinovirus 3C-protease (HRV-3C) cleavable N-terminal 6-His affinity tag. DNA encoding amino acids 448–703 of Smc6 was cloned into pET-52b(+), which express the recombinant protein with an N-terminal HRV-3C cleavable Strep$_{II}$ affinity tag (Merck Millipore).

**Expression and purification.** *E. coli* strain Rosetta2(DE3) (Merck Millipore) was co-transformed with pCDF-Smc5 and pET52-Smc6 plasmids. Transformants were then selected on LB-agar plates supplemented with antibiotics. From an overnight culture, 25 ml was used to inoculate a 2 l flask, containing 1 l of Turbo-broth media (Molecular Dimensions, Newmarket, UK) again supplemented with antibiotics. Cultures were grown in an orbital-shaking incubator, at 37 °C, until an optical density of 1.5 units at a wavelength of 600 nm was reached. The temperature was then reduced to 18 °C, and recombinant protein expression induced by the addition of 0.1 M isopropyl β-D-1-thiogalactopyranoside. Cells were subsequently harvested by centrifugation after 16 h at the reduced temperature. The resultant pellet was stored at −20 °C until required.

The cell pellet resulting from 4 l of culture was resuspended in Buffer A (50 mM HEPES.NaOH pH 7.5, 250 mM NaCl, 10 mM imidazole, 0.5 mM TCEP) supplemented with protease inhibitor tablets (Roche, Burgess Hill, UK). Cells were then disrupted by sonication, and insoluble material removed by centrifugation. The resultant supernatant was incubated with Talon resin (TaKaRa Bio, Saint-Germain-en-Laye, France) pre-equilibrated in Buffer A. After successive washes with Buffer A to remove unbound material, the retained recombinant proteins were eluted by the addition of Buffer B (50 mM HEPES.NaOH pH 7.5, 250 mM NaCl, 300 mM imidazole, 0.5 mM TCEP). This eluate was then loaded onto a *Strep*-Tactin Superflow Plus 5 ml cartridge (Qiagen, Germantown, USA) pre-equilibrated with Buffer C (20 mM HEPES.NaOH pH 7.5, 250 mM NaCl, 0.5 mM TCEP). After successive washes with Buffer C to remove unbound material, the retained recombinant proteins were eluted with Buffer C supplemented with 2 mM D-desthiobiotin. The affinity tags were then cleaved by overnight incubation with human rhinovirus 3C-protease at 4 °C. The proteins were concentrated to a final volume of 5 ml using Vivaspin 20 (10,000 MWCO) centrifugal concentrators (Sartorius Stedim Biotech, Goettingen, Germany) and then loaded onto a Superdex 200 size exclusion chromatography column (GE Healthcare Life Sciences, Little Chalfont, UK) pre-equilibrated with Buffer C as the final purification step. Fractions containing the purified complex were identified by SDS-PAGE, pooled and then concentrated to 12 mg ml$^{-1}$ and either used immediately or flash-frozen in liquid N$_2$ and stored at −80 °C until required.

**Selenomethionine incorporation.** Selenomethionine-labelled protein was made using commercial media and protocols; SelenoMet Medium Base plus Nutrient Mix and selenomethinone solution (Molecular Dimensions, Newmarket, UK).

**Crystallography.** Smc5/6-hinge was crystallized in MRC2 crystallization plates at 18 °C, using the sitting drop vapour diffusion method; mixing 200 nl of protein (6 mg ml$^{-1}$) with 200 nl of precipitant (100 mM Bis-Tris pH 7.5, 2.1 M ammonium sulphate), which was allowed to equilibrate against a well containing 50 μl of precipitant. Crystals typically appeared after 2–3 days.

Micro-seeding was generally required in order to generate single crystals. Cryoprotection for data collection was achieved by step-wise soaking of crystals in buffers containing increasing amounts of sucrose, to a final concentration of 30% (w/v).

Diffraction data were collected to 2.75 Å at 100 K on station I04 at the Diamond Light Source (DLS), Didcot, UK. Data were processed with XDSgui (http://strucbio.biologie.uni-konstanz.de/xdswiki/index.php/XDSGUI) and XDS[53] and then scaled using Aimless[54], a package included with the CCP4 software suite[55].

Phases were calculated from a single-wavelength anomalous dispersion experiment measured at 0.9788 Å (absorption edge of Se, as determined by fluorescence scan) on DLS station I24, using crystals containing selenomethionine-derivitized protein. The position of 26 Se atoms, at a resolution of 4.7 Å, could be determined using the AutoSol pipeline of the PHENIX software suite[56–59]. Refinement of heavy atom position, density modification and phase extension with the native dataset was also carried out with AutoSol. An initial model was produced by phenix.autobuild[56], which was extended and improved by iterative rounds of manual building in Coot[60] and refinement with BUSTER[61] and phenix.refine[56] to produce the final working model. Crystallization statistics are reported in Supplementary Table 2.

**Experiments in yeast.** The yeast strains used in this study are listed in Supplementary Table 3 and cultured under standard conditions[62]. Base strains where the locus and a selectable ura4 marker are flanked by incompatible loxP and loxM sites were constructed, as in ref. 34, in order to facilitate rapid integration of smc5 and smc6 constructs by recombination cassette exchange. Mutant alleles were constructed by site-directed mutagenesis of the wild-type gene in the plasmid pAW8, and propagated in the E. coli strain DH5α. Cassette exchange was then performed as described[34]. A wild type construct was always transformed in parallel to control for both transformation and integration efficiencies, especially when the introduced mutation(s) produce a lethal phenotype (as these result in no viable integrants). Where no viable integrants were obtained, lethality was confirmed by sporulation, following integration into a diploid base strain. The resulting mutant strains were confirmed by PCR and sequencing. For DNA damage sensitivity assays, 10-fold serial dilutions of cells were spotted on the appropriate plates, and then grown at the indicated temperature. Images were taken after 3 days. In assays, mutant strains were compared to both wild-type S. pombe and to base strains where the ura4 marker had been replaced with a wild type copy of the gene, in order to control for the presence of the lox sites.

**Experiments in human cells.** 2 μg of pTRE3G (TakaraBio) containing a doxycycline-inducible construct expressing eGFP-fused wild-type or mutant human Smc5, or an eGFP vector control, were co-transfected with pCI-puro (Promega) in a 2:1 ratio, into U2OS cells (ATCC, LGC Standards) containing the Tet-On 3G transactivator protein grown in tetracycline-free media (PanBioTech) on 10 cm dishes, using 9 μl GeneJuice transfection reagent (Merck Millipore). The culture medium was replaced 12–18 h after transfection, and the cells incubated for a further 14 days under puromycin and G418 selection, before expression of Smc5 was induced by the addition of doxycycline (1 μg ml$^{-1}$). Cell viability was scored after 48 h using a trypan-blue cell viability assay. Error bars represent 1 standard deviation of the mean, for three independent experiments carried out in triplicate.

Endogenous Smc5 was knocked down using Dharmacon siRNA smartpool (L-014117-01-005); scrambled control (D-001810-01-05).

**Antibodies.** *Primary.*

*His*-affinity tag, mouse monoclonal at 1:5,000 dilution (631212, Takara Bio).
*Strep*-affinity tag, mouse monoclonal at 1:5,000 dilution (34850, Qiagen).
Human Smc5, rabbit polyclonal at 1:500 dilution, ref. 63.
S. pombe Smc5 (Spr18), sheep polyclonal at 1:500 dilution, ref. 38.
S. pombe Smc6 (Rad18), rabbit polyclonal at 1:500 dilution, ref. 23.
Alpha-tubulin, mouse monoclonal at 1:20,000 dilution (T5168, Sigma-Aldrich).
*Secondary.*
Anti-mouse IgG HRP-conjugate, sheep polyclonal at 1:10,000 dilution (NA931, GE Healthcare).
Anti-rabbit HRP-conjugate, swine polyclonal at 1:10,000 dilution (P0217, Agilent Technologies).
Anti-sheep HRP-conjugate, rabbit polyclonal at 1:4,000 dilution (P0163, Agilent Technologies).
Anti-mouse HRP-conjugate, rabbit polyclonal at 1:4,000 dilution (P0161, Agilent Technologies).

**Biochemical and biophysical experiments.** All biochemical and biophysical experiments used extended-hinge Smc5/6 expression constructs: DNA encoding amino acids 364-692 of Smc5 cloned as above; DNA encoding amino acids 462-773 of Smc6 cloned as above.

**Hinge stability assays.** Hinge stability assays followed a similar protocol to that previously reported by ref. 39. Briefly, E. coli lysates containing co-expressed recombinant His-tagged Smc5-hinge and Strep$_{II}$-tagged Smc6-hinge were incubated/rotated with ~250 μl of Talon resin (Takara Bio) pre-equilibrated in assay buffer: 50 mM HEPES.NaOH pH7.5, 1,000 mM NaCl, 0.5 mM TCEP. After successive washes with assay buffer to remove any unbound material, samples were analysed for the presence of both proteins by SDS-PAGE/western blot.

**Size exclusion chromatography.** Smc5/6-hinges were applied to a Sephadex 200 10/300 GL size exclusion column (GE Healthcare) pre-equilibrated in 20 mM

HEPES.NaOH pH7.5, 250 mM NaCl, 0.5 mM TCEP. The column was calibrated using a Gel Filtration Calibration Kit LMW (28-4038-41 GE Healthcare).

**Small angle X-ray scattering.** Data were recorded at on BioSAXS beamline BM29 (ESRF, Grenoble, France) or B21 (Diamond Light Source, Didcot, UK). The experimental setup included an in-line HPLC connected to a Sephadex 200 5/150 GL size exclusion column. Experimental two-dimensional data were reduced to a one-dimensional scattering profiles by in-house software[64]. Data were averaged and scaled using ScÅtter[46]. ScÅtter was also used to calculate $P(r)$-distributions from deposited PDB coordinates. Programmes of the ATSAS software package were used to generate *ab initio*, single phase, dummy atom models[65,66].

***Ab initio* dummy atom models.** The pair-distance distribution function calculated for Smc5/6-hinge was used as the input to DAMMIF, a program for rapid *ab initio* shape determination in small-angle scattering[66]. Fifty independent dummy atom models were obtained by running the program in 'slow' mode. Subsequently, DAMCLUST was used to cluster the models into groups, and then to identify individual models that best represented each grouping[65]. Analysis with FoXS[44] (Fast SAXS Profile Computation with Debye Formula[44]) indicated that a single representative dummy atom model, from the largest cluster of 27 related models (deviation 0.61), provided the best overall fit to the experimental scattering data; χ value = 0.47.

**Oligonucleotides.** Purified DNA oligonucleotides were purchased from Eurofins Genomics, Ebersberg, Germany. FLU = Fluorescein. BIO = Biotin.
A: 5′-TTAGTTGTTCGTAGTGCTCGTCTGGCTCTGGATTACCCGC-FLU-3′
B: 5′-GCGGGTAATCCAGAGCCAGACGAGCACTACGAACAACTAA-3′
10: 5′-FLU or BIO-GCTCGTCTGG-3′
15: 5′-FLU or BIO-GCTCGTCTGGCTCTG-3′
20: 5′-FLU or BIO-CTCGTCTGGCTCTGGATTAC- 3′
25: 5′-FLU or BIO-GCTCGTCTGGCTCTGGATTACCCGC-3′
30: 5′-FLU or BIO-GTAGTGCTCGTCTGGCTCTGGATTACCCGC-3′
35: 5′-FLU or BIO-GCGGGTAATCCAGAGCCAGACGAGCACTACGAACG-3′
40: 5′-FLU or BIO-GCGGGTAATCCAGAGCCAGACGAGCACTACGAAC
AACTAA–3′

**EMSA.** Oligonucleotide A (ssDNA) or annealed A + B oligonucleotides (dsDNA) at a concentration of 100 nM, were mixed with increasing concentrations of wild-type Smc5/6-hinge, in 20 mM HEPES.NaOH pH 7.5, 100 mM NaCl, 1 mM EDTA, 0.5 mM TCEP, and incubated for 10 min at room temperature. Samples were then analysed on 6% v/v native polyacrylamide gels (6% DNA Retardation Gel, ThermoFisher Scientific) containing 0.5X tris-borate-EDTA (TBE) and visualized by direct scanning of the gel in a Fuji FLA-5100 Fluorescent Image Analyser.

**Pull-down with biotinylated oligonucleotides.** Highly saturating concentrations of biotinylated oligonucleotide (100 μl at 100 μM) were incubated with 50 μl of Streptavidin Mag Sepharose resin (GE Healthcare) in 20 mM HEPES.NaOH pH7.5, 100 mM NaCl, 1 mM EDTA, 0.5 mM TCEP. After incubation, any unretained oligonucleotide was removed by sequential washes of the resin in the same buffer. The resin was then incubated with 120 μl of wild-type Smc5/60-hinge (1.5 mg ml$^{-1}$) for a period of 30 min, at 4 °C, with agitation. Unretained protein was removed by sequential buffer washes. Retained material was analysed by standard SDS-PAGE with colloidal Coomassie blue staining.

**Fluorescence polarization.** Oligonucleotides at a concentration of 100 nM were incubated with increasing concentrations of wild-type Smc5/6-hinge, in 20 mM HEPES.NaOH pH7.5, 100 mM NaCl, 1 mM EDTA, 0.5 mM TCEP, and incubated for 10 min at room temperature. Fluorescence polarization was measured in a POLAR-star OMEGA multimode plate reader (BMG Labtech GmbH, Offenburg, Germany).

**Determination of K$_d$.** Binding data were analysed with GraphPad Prism 6.0, by non-linear fitting with a one-site binding model, to give the reported dissociation constants ($K_d$). All data represent the mean of three separate experiments, and error bars represent 1 standard deviation.

**SwitchSENSE.** 5′-HS-(CH$_2$)$_6$-GTGTGAACCCTCCAACAAAGGTAGCATTTGC CAGCTCTCGTGATGCAG-Cy3-3′ Cy3-fluorescently labelled oligonucleotide was immobilized to the surface, via thiol chemistry, to the gold microelectrodes of one switchSENSE BioChip. HS-(CH2)6 = Thiolated linker, Cy3 = indocarbocyanine. The methodology employed in switchSENSE technology is described in ref. 67. Experiments were carried out on a DRX 2400 instrument. Binding rate constants were determined using the supplied data analysis software package, using experimental data from F$_{down}$ kinetic experiments.

**Gels and western blots.** Uncropped scans of blots and gels are provided in Supplementary Fig. 9.

**Data availability.** Coordinates and structure factors have been deposited in the RSCB Protein Data Bank, with accession code (5MG8). All other relevant data are available from the corresponding authors.

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

## Acknowledgements

Mark Roe for assistance with X-ray diffraction data collection. Diamond Light Source Ltd., (DLS) Didcot, UK, for continuing access to synchrotron radiation. Robert Rambo for rapid access to B21 at DLS and Katsuaki Inoue for help and assistance. Suzanne Vidot for construction of the *smc6* base-strain. Alessandro Bianchi and Stuart Rulten, for help and assistance with the human cell experiments. Duncan Borthwick, Dynamic Biosensors GmbH, for access to switchSENSE technology. Members of the Pearl/Oliver, Murray and Carr laboratories for helpful discussion. Supported by: Cancer Research UK Programme Grant C302/A14532 (A.W.O., L.H.P.), MRC Project Grants G1001668 (A.W.O., J.M.M., A.R.L., L.H.P.), G0901011 (J.M.M., A.R.L.), G1100074 (O.S.W.).

## Author contributions

Conceptualization: L.H.P., A.R.L., J.M.M., A.W.O.; Methodology: L.H.P., J.M.M., A.W.O.; Investigation: A.A., H.Q.D., O.S.W., L.M.P., M.A.S., G.A.M., T.W., A.W.O.; Original draft: A.W.O.; Writing, review and editing: A.R.L., L.H.P., J.M.M., A.W.O.; Visualization: A.W.O.; Supervision: L.H.P., J.M.M., A.W.O. Funding acquisition: A.W.O., J.M.M., A.R.L., L.H.P.

## Additional information

**Competing financial interests:** The authors declare no competing financial interests.

