## [Peer Review File · Nature Communications]

Reviewers' Comments:

Reviewer #1 (Remarks to the Author):

Alt et al present the crystal structure of the Smc5-Smc6 "hinge" domain that is required to dimerise the full Smc5/6 complex. Structural features unique to Smc5/6 are identified and an extensive analysis of the dimerisation and DNA-binding properties of the complex undertaken. A set of mutants that compromise these activities are shown to affect cell survival and/or DNA repair functionality. Smc5/6 are arguably the least well understood members of the Smc family, and so this work represents a useful contribution to both our understanding of both SMC proteins and DNA repair mechanisms. The relationship of DNA-binding to in vivo phenotype is important, though mechanistic understanding is still lacking, but perhaps beyond the scope of this work. There are also multiple Smc hinge structures described in the literature, so the principle novelty is in the details of the dimerisation and DNA-binding activity. The structural and biochemical data presented in the paper appear solid and conclusions are in general not over-speculative.

Specific points:

1. The manuscript would obviously be considerably strengthened if the hinge could be co-crystallised with DNA. I assume the authors attempted this with negative results, but was wondering if there are any specific issues that would preclude this?
2. It's rather odd that the S610G mutant shows differential effects with different DNA-damaging agents (e.g. no phenotype with MMS or U.V.). While the induced upstream signal may differ, one would expect the mechanism of Smc5/6 during repair to be consistent. Can the authors explain these differences?
3. The experiments in human cells are fairly brief, with cell-death being the only read-out. Some more detailed examination of terminal phenotype would have been interesting.
4. Phe528 is proposed to close the arms of the hinge and promote the tilted conformation of the hinge. Given that the mutant has already been made for DNA-binding analyses (and thus is presumably stable and soluble), the effect on the conformation of the hinge-coiled-coil could be tested by SAXS. Alternatively the speculation should be removed.

Minor comments:

1. None of the main figures have titles and most of the supplementary ones are clipped (at least in my PDF). This is a bit irritating.
2. The "linker" domain in Figure 1b is not very clear, particularly in Smc5 (grey on pale blue).
3. Line 159 - "nexus" could be replaced with a word more often encountered in common English usage.

Reviewer #2 (Remarks to the Author):

The manuscript by Alt et al describes structure and functional analyses of the hinge regions of the Smc5 and 6 proteins from the fission yeast. These hinge regions form a toroidal structure similar to those seen in cohesin and condensin, but with distinctive subunit interface. A "molecular latch" - composed of the long loop C from Smc5 - provides an interface with Smc6, in addition to the classical SMC hinge North and South interfaces. The loop C contacts both Subdomain II and the linker between two subdomains of Smc6 hinge, with S610 and Y612 being highly conserved. Mutating both sites is lethal, and individual mutations lead to DNA damage sensitivity and slow growth (in the case of Y612G). Equivalent mutations were shown to cause similar defects in human cells. The Smc6 hinge also contains two glycines as seen in other SMC proteins. Another mutation F528A in Smc6 causes DNA damage sensitivity. A couple of other mutations were also examined. Moreover, the authors showed that Smc5/6 hinge prefers binding to ssDNA over dsDNA of 15-45nt. Mutating a pair of arginines of Smc5 (R609/R615E), which binds to a sulphate ion in the structure, leads to reduction of DNA binding and DNA damage sensitivity. Finally SAXS tests suggest that Smc5/6 hinge complex adopts an arm-closed conformation with the coiled coil region mostly parallel to each other.

The biophysical analyses of the Smc5/6 hinges included in this manuscript are the first to describe shared and unique features of the hinge regions of the Smc5/6 complex. Though DNA binding and SAXS analyses are somewhat preliminary, they provide useful information regarding how Smc5/6 hinges could engage DNA and connect with arms. Mutations disrupting specific features, examined in vivo, in vitro, or both, largely verified predictions from the structure. To further substantiate their conclusions and to avoid some confusions, I would suggest the authors to address a few issues. For example, structural information was not presented fully, with several pieces of information missing. In addition, a few key mutations need to be examined both in vivo and in vitro to fully assess how they cause a change in proteins'

functions. A few other points are also indicated below.

1. The Y612G mutation causes a strong defect in yeast cells, though the mechanism is not clear. Based on structural information, Y612G could impair hinge interaction. However, no data were provided for this mutation, rather Y612A was examined for interaction. The authors should examine Y612G for hinge interaction *in vitro*, and for the integrity of the Smc5/6 complex *in vivo*. This would allow correlation between biochemical and genetic findings. Alternatively, the Y612A mutant should be examined in yeast cells. Along the same line, Smc6-F5828A should be tested for alteration of biochemical features to understand why it only causes damage sensitivity but not growth.

2. The manuscript is generally clear, however certain parts need better explanation and clarification. For example, several constructs used in the study were not described in the text, causing confusion. What "long arm" or "the region of core folds" constructs refer to? They should be defined by amino acid coordinates in the text when first appear. In addition, the rationale for changing constructs should be provided. In Page 3, amino acid coordinates of the fragments used in structural determination and those shown in Figure 1B need to be clearly indicated.

3. The complete sequence seen in the structure should be aligned with 2nd structure and be included as a panel in figure 1 or 2. This panel should also include the denotation of Subdomain I and II, in order for readers to follow the structural information easily. Besides the loops, which are assigned as Loop A-C, the alpha helices and beta-sheets should also be properly numbered. The region that forms coiled coil should be included. Critical residues examined in the studies should be labeled and highlighted. The alignment should also include the similar regions of the Smc5 and 6 proteins from other species.

4. Though the classical north and south interfaces between Smc5/6 hinges are not unique, it is important to clearly define the regions on each protein that constitute these two interfaces, both in text and as a figure panel. A comparison with the North and South interfaces of other SMC hinge regions will be useful. The diameter of the central cavity of the ring-shaped structure and information regarding the conservation at these interfaces should be provided.

5. The rationale for generating 6-mut (S692E, G694K, and S696E) is not clear. As G683 and G694 of Smc6 are conserved, one would assume that their mutations should be made, instead of 6-mut. Why the authors consider S692 and S696 to be important - are they highly conserved? In addition, changing from E or K can be dramatic, and milder mutations should be used. More interface mutations should be tested in yeast cells.

6. Protein levels of yeast mutations should be examined to rule out the possibility that the

mutations simply reduce protein levels. In addition affinity between the two hinge domains should be shown

7. Smc5-R609E/R615E should be tested for interaction with Smc6 hinge to discern whether its reduced DNA binding is due to impairment of Smc6 vs. DNA association. Also, Smc6-Y612A reduces hinge-hinge interaction but maintain DNA binding, raising the possibility that ssDNA binding is mainly conferred by a single Smc subunit. The authors should examine this possibility to be able to accurately assess Smc5/6 interact with DNA.

8. Smc5/6 has been suggested to be similar to prokaryotic SMCs, the authors should discuss this view in light of their structural data. Why the authors think that the Smc5/6 hinge can be more dynamic in loading/unloading based on the molecular latch in lieu of glycine motifs in Smc5? It appears that the latch might provide more stability. Please provide a clear rationale for this argument.

Reviewer #3 (Remarks to the Author):

Multi-subunit Smc complexes fulfill diverse functions in genome maintenance. The Smc hinge domain is a central element of Smc, which dimerizes two Smc proteins to create Smc heterodimers (cohesin Smc1-3, condensin Smc2-4, and Smc5-6). This manuscript by the Oliver, Murray and Pearl labs elucidates for the first time the structure of the Smc5-6 hinge. As expected from low but significant levels of sequence similarity to other Smc hinge domains, the Smc5-6 hinge in the *S. pombe* crystal structure displays a typical Smc hinge fold. However, some clear deviations from the otherwise highly conserved architecture are obvious in Smc5-6. The authors describe these structural differences in detail. Using extensive mutagenesis, they assign functions to the newly identified structural elements: An exposed loop on Smc5 is shown to contribute to the dimerization interfaces. Consistently, it is found to be important for the survival in the presence of DNA damaging drugs. A loop on Smc6 is proposed to anchor the coiled coil onto the Smc5-6 hinge, while several arginine residues located in a patch of positively charged surface residues are being implicated in the preferential binding of the Smc5-6 hinge to single-stranded DNA. Small angle X ray scattering supports the existence of the crystallographically observed closed conformation of the Smc5-6 hinge dimer (and is inconsistent with an open form). This finding extends a recently proposed concept for condensin to the Smc5-6 complex. However, the functional implications are somewhat unclear.

Altogether, the report will be highly valuable for the field of Smc and chromosome biology. The manuscript is extensive but easy to follow. The data is well presented and generally makes a very solid impression. The conclusions from the work are valid. The manuscript should thus be a

good candidate for publication in Nature Communications.

Specific comments

The R609E/R615E double mutant of Smc5 shows a strong growth defect, drug sensitivity and a DNA binding phenotype. Since both residues are located close to the dimerization interface, it is important to exclude the possibility that the observed effects are not indirectly caused by a dimerization defect. Please include the mutant in the pull-down assay (shown in Fig. 4D) or perform SAXS measurements.

In the light of obvious similarity between the hinge-coiled coil arrangement (Fig. 7) in the new Smc5-6 hinge structure and in published Smc2-4 hinge structures, it is surprising that the Smc6 coiled coil (not Smc5 - the presumed homolog of Smc4) contacts the hinge domain. Could the authors speculate on whether this is due to convergent evolution (of hinge-coiled coil contacts) or whether Smc5 and Smc6 may possibly have been wrongly assigned to the sub-families of SMC proteins?

Please include quality control experiments for the anti-Smc5 antibody used in Fig. 3B (the expression levels of exogenous protein seems surprisingly low compared to endogenous Smc5 despite the strong phenotype seen in Fig. 3C). Is the Smc5 mutant dominant in human cells but not yeast? If so, why?

It would be highly desirable to have different names (such as A and A' for loops and residues) and different color schemes on Smc5 and Smc6 throughout the manuscript. Currently, it is difficult to attribute residues and features to Smc5 or Smc6 hinge domains in some figures.

The title to Figure 7 is misleading since only a single conformation has been detected for Smc5-6 so far. Please modify accordingly.

Reviewer #1 (Remarks to the Author):

Alt et al present the crystal structure of the Smc5-Smc6 "hinge" domain that is required to dimerise the full Smc5/6 complex. Structural features unique to Smc5/6 are identified and an extensive analysis of the dimerisation and DNA-binding properties of the complex undertaken. A set of mutants that compromise these activities are shown to affect cell survival and/or DNA repair functionality. Smc5/6 are arguably the least well understood members of the Smc family, and so this work represents a useful contribution to both our understanding of both SMC proteins and DNA repair mechanisms. The relationship of DNA-binding to in vivo phenotype is important, though mechanistic understanding is still lacking, but perhaps beyond the scope of this work. There are also multiple Smc hinge structures described in the literature, so the principle novelty is in the details of the dimerisation and DNA-binding activity. The structural and biochemical data presented in the paper appear solid and conclusions are in general not over-speculative.

Specific points:

1. The manuscript would obviously be considerably strengthened if the hinge could be co-crystallised with DNA. I assume the authors attempted this with negative results, but was wondering if there are any specific issues that would preclude this?

We can confirm that we have tried co-crystallising the hinge with DNA, but have not been successful to date. However, these experiments are still ongoing in the laboratory and we continue to travel optimistically.

It may be that the packing of hinge molecules in our crystals is not compatible with DNA-binding; something we are currently investigating, including attempts at 'crystal engineering' to alter crystal packing.

Furthermore, obtaining a consistent register for any bound ssDNA is likely to be problematic, as we don't expect the hinge to have any sequence-specificity. We are therefore currently exploring different types of DNA containing short sections of secondary structure, and/or capping of ends with biotin etc., in order to promote a single bound conformation, which is then amenable to crystallisation.

2. It's rather odd that the S610G mutant shows differential effects with different DNA-damaging agents (e.g. no phenotype with MMS or U.V.). While the induced upstream signal may differ, one would expect the mechanism of Smc5/6 during repair to be consistent. Can the authors explain these differences?

We thank the reviewer for highlighting this matter. On re-inspection of the submitted manuscript, it appears that we have inadvertently included incorrect spot-test panels in some places. We apologise for this error. We have now carefully re-examined each image, and can verify that they are all now correct.

Affected panels: Figure 3A, 0.005% MMS; Figure 5F, 0.005% MMS and 50 + 100 J/m² UV.

As expected/suggested S610G is demonstrably sensitive to all agents tested (CPT, MMS, HU, UV) but with a much weaker phenotype compared to the Y612G mutation (as reported). We have altered the manuscript to now read: 'Mutation of Smc5-Ser610 to glycine (S610G) produced a weaker phenotype, with some sensitivity to all agents tested'. Page 5; Lines 120-123.

3. The experiments in human cells are fairly brief, with cell-death being the only read-out. Some more detailed examination of terminal phenotype would have been interesting.

This experiment was designed as a relatively simple way of examining if mutation of the conserved latch feature (Y626G) produced any discernable phenotype in human cells.

We agree that a more detailed examination of the mode / means of cell death would be of interest, but would argue that this sits somewhat outside of the scope of this manuscript.

4. Phe528 is proposed to close the arms of the hinge and promote the tilted conformation of the hinge. Given that the mutant has already been made for DNA-binding analyses (and thus is presumably stable and soluble), the effect on the conformation of the hinge-coiled-coil could be tested by SAXS. Alternatively the speculation should be removed.

As requested, we have measured the SAXS profile of F528A (as well as R609E/R615E). This data is now included in a new supplementary figure (Supp. Fig. 8).

The scattering profiles of WT, R609E/R615E and F528A are essentially identical; indicating that both sets of mutations do not, in themselves, produce a gross change in the conformation of the Smc5/6-hinge – but still alter ssDNA binding behavior in vitro, and produce strong phenotypes in vivo.

As requested, we have removed several of the more speculative statements relating to the F528A mutation (see below) and in light of the new observations amended the discussion (see revised manuscript).

- *Page 11, end of 4th paragraph: “serving to ‘close’ the hinge and promote arm association, and resulting in the observed ‘tilted-hinge’ conformation”*
- *Page 13; end of 3rd paragraph: “supportive of the concept that arm architecture (open or closed) regulates access to the DNA-binding residues of interaction point #2 at the Smc6-hub”*
- *Page 13, start of 4th paragraph: “together then these data indicate that the conformational state of the hinge is critically important and directly coupled to DNA-binding capability”*

Minor comments:

1. None of the main figures have titles and most of the supplementary ones are clipped (at least in my PDF). This is a bit irritating.

We believe that these were erroneously removed during the PDF conversion process. We apologise for the inconvenience that this has caused. We will note this issue in any future submissions.

2. The "linker" domain in Figure 1b is not very clear, particularly in Smc5 (grey on pale blue).

As requested, we have altered the colour scheme in Fig1B, in order to make the linker regions more visible (also see response to Reviewer 3 below).

3. Line 159 - "nexus" could be replaced with a word more often encountered in common English usage.

On this occasion, we would prefer to retain the word nexus, as its meaning is highly specific and fully appropriate to the context it is used in: “a connection or series of connections linking two or more things”, or “a central or focal point”.

Reviewer #2 (Remarks to the Author):

The manuscript by Alt et al describes structure and functional analyses of the hinge regions of the Smc5 and 6 proteins from the fission yeast. These hinge regions form a toroidal structure similar to those seen in cohesin and condensin, but with distinctive subunit interface. A "molecular latch" - composed of the long loop C from Smc5 - provides an interface with Smc6, in addition to the classical SMC hinge North and South interfaces. The loop C contacts both Subdomain II and the linker between two subdomains of Smc6 hinge, with S610 and Y612 being highly conserved. Mutating both sites is lethal, and individual mutations lead to DNA damage sensitivity and slow growth (in the case of Y612G). Equivalent mutations were shown to cause similar defects in human cells. The Smc6 hinge also contains two glycines as seen in other SMC proteins. Another mutation F528A in Smc6 causes DNA damage sensitivity. A couple of other mutations were also examined. Moreover, the

authors showed that Smc5/6 hinge prefers binding to ssDNA over dsDNA of 15-45nt. Mutating a pair of arginines of Smc5 (R609/R615E), which binds to a sulphate ion in the structure, leads to reduction of DNA binding and DNA damage sensitivity. Finally SAXS tests suggest that Smc5/6 hinge complex adopts an arm-closed conformation with the coiled coil region mostly parallel to each other.

The biophysical analyses of the Smc5/6 hinges included in this manuscript are the first to describe shared and unique features of the hinge regions of the Smc5/6 complex. Though DNA binding and SAXS analyses are somewhat preliminary, they provide useful information regarding how Smc5/6 hinges could engage DNA and connect with arms. Mutations disrupting specific features, examined in vivo, in vitro, or both, largely verified predictions from the structure. To further substantiate their conclusions and to avoid some confusions, I would suggest the authors to address a few issues. For example, structural information was not presented fully, with several pieces of information missing. In addition, a few key mutations need to be examined both in vivo and in vitro to fully assess how they cause a change in proteins' functions. A few other points are also indicated below.

1. The Y612G mutation causes a strong defect in yeast cells, though the mechanism is not clear. Based on structural information, Y612G could impair hinge interaction. However, no data were provided for this mutation, rather Y612A was examined for interaction. The authors should examine Y612G for hinge interaction in vitro, and for the integrity of the Smc5/6 complex in vivo. This would allow correlation between biochemical and genetic findings. Alternatively, the Y612A mutant should be examined in yeast cells.

Based on our knowledge of structural biology, in this particular context, mutation of Y612 to either alanine or glycine (Y612A / G) would have the same disruptive effect on hinge-dimerisation, i.e. by the removal of the phenol moiety of the tyrosine side-chain, and its associated packing interactions with Smc6-Phe611 and Smc6-Tyr613, as well as the hydrogen-bonds with Smc6-Glu647 and Smc6-Lys648 (Figure 2B).

Along the same line, Smc6-F5828A should be tested for alteration of biochemical features to understand why it only causes damage sensitivity but not growth.

We confirm by co-expression / co-purification experiment that the F528A mutant does not disrupt hinge-dimerisation. This data is presented in a new panel (B) of Supplementary Figure 5. Similarly, the SAXS profile of F528A demonstrates that the protein is fully folded, and adopts the same conformation in solution as the wild-type protein (see response to Reviewer 1, point #4).

2. The manuscript is generally clear, however certain parts need better explanation and clarification. For example, several constructs used in the study were not described in the text, causing confusion. What "long arm" or "the region of core folds" constructs refer to? They should be defined by amino acid coordinates in the text when first appear. In addition, the rationale for changing constructs should be provided. In Page 3, amino acid coordinates of the fragments used in structural determination and those shown in Figure 1B need to be clearly indicated.

We have amended the first two paragraphs of the results section (bottom page 3, top page 4) to reference a new supplementary table (Table S1) which clearly defines the amino acid boundaries of each construct used.

In the manuscript itself, we have included a definition of the amino acid boundaries of the 'core fold' of the Smc5/6-hinge (Smc5: amino acids 434-634, Smc6: 524-711).

We have revised the terms 'long-arm' and 'short-arm' (where used) to read 'extended-hinge' and 'truncated-hinge' respectively; to keep a more consistent nomenclature throughout the manuscript, and which should assist the reader.

Note: we only use the 'truncated-hinge' constructs in a single experimental set (analytical size exclusion chromatography) and the justification for their use was already provided in the associated figure legend (Figure 5C).

3. The complete sequence seen in the structure should be aligned with 2nd structure and be included as a panel in figure 1 or 2. This panel should also include the denotation of Subdomain I and II, in

order for readers to follow the structural information easily. Besides the loops, which are assigned as Loop A-C, the alpha helices and beta-sheets should also be properly numbered. The region that forms coiled coil should be included. Critical residues examined in the studies should be labeled and highlighted. The alignment should also include the similar regions of the Smc5 and 6 proteins from other species.

We have modified Figure 1C to define and show the boundaries of Subdomain I, II and the linker regions. We have also created a new subsidiary panel in Supplementary Figure 1 (panel C), in order to more fully provide the various details requested by the reviewer.

4. Though the classical north and south interfaces between Smc5/6 hinges are not unique, it is important to clearly define the regions on each protein that constitute these two interfaces, both in text and as a figure panel. A comparison with the North and South interfaces of other SMC hinge regions will be useful. The diameter of the central cavity of the ring-shaped structure and information regarding the conservation at these interfaces should be provided.

We have modified Figure 4C to indicate the amino acid extents of the beta-strands that pair to produce the two beta-sheets at the North and South interfaces. We believe that an extensive comparison of North and South interfaces between SMC hinges would indeed be interesting, but again would argue that this sits somewhat outside of the scope of this manuscript.

We have amended Figure 3D to indicate the (estimated) diameter of the central cavity.

5. The rationale for generating 6-mut (S692E, G694K, and S696E) is not clear. As G683 and G694 of Smc6 are conserved, one would assume that their mutations should be made, instead of 6-mut. Why the authors consider S692 and S696 to be important - are they highly conserved? In addition, changing from E or K can be dramatic, and milder mutations should be used. More interface mutations should be tested in yeast cells.

On revisiting the manuscript, we can understand where confusion may arise. 6-mut was specifically designed to ensure complete disruption of the South interface, and simply acts as a control for the important observation that a single-point mutation of Y612 is sufficient to perturb/break the North interface. As it only serves in this function, we do not think that generation of "milder" mutations is warranted in this instance.

We have amended the text of the manuscript to more clearly illustrate our experimental path:

Hinge-domain association

*We hypothesised that mutation of Smc5-Loop C would specifically disrupt the North interface. To provide a suitable control, we also generated a series of mutations along the last β -strand of Smc6 (centred around the conserved glycine residue Gly694), designed to sterically disrupt the South interface (6-Mut: Smc6-S692E, -G694K, -S696E; see **Supplementary Fig. 4** for additional details).*

In order to assist the reader, we have also separated the previous subheading "Conserved glycine motifs and hinge-domain association" into two separate parts, i.e. "Conserved glycine motifs" and "Hinge-domain association" in order to demarcate a clear separation between the two sections.

We have also labelled Gly683 and Gly694 in Figure 4C, as well as in Supplementary Figure 4.

6. Protein levels of yeast mutations should be examined to rule out the possibility that the mutations simply reduce protein levels. In addition affinity between the two hinge domains should be shown

As requested, we have confirmed expression levels for each mutant by western blot against yeast whole cell extracts (Trichloroacetic acid protein precipitation). These are presented in a new panel (A) forming part of Supplementary Figure 3.

7. Smc5-R609E/R615E should be tested for interaction with Smc6 hinge to discern whether its reduced DNA binding is due to impairment of Smc6 vs. DNA association.

As requested, we confirm that the mutations do not affect hetero-dimerisation or conformation in solution; see: co-expression / co-purification experiment for Smc5-R609E/R615E (new panel B in Supplementary Figure 5); and SAXS profile (new Supplementary Figure 8).

Also, Smc6-Y612A reduces hinge-hinge interaction but maintain DNA binding, raising the possibility that ssDNA binding is mainly conferred by a single Smc subunit. The authors should examine this possibility to be able to accurately assess Smc5/6 interact with DNA.

Our data clearly demonstrates that interactions with ssDNA occur with BOTH Smc5 AND Smc6; but in a hierarchical manner (see Supplementary Data 6, panel B).

- *Mutation of Smc5-R609E/R615E (DNA interaction point #1) abrogates all ssDNA interactions (with short and long ssDNA)*
- *Mutation of Smc6-R706C strongly affects interaction with longer stretches of ssDNA.*
- *Mutation of Smc6-F528A changes ssDNA interaction behavior with longer stretches of ssDNA.*

8. Smc5/6 has been suggested to be similar to prokaryotic SMCs, the authors should discuss this view in light of their structural data.

Whilst this is a valid and interesting area of research, we feel that this is somewhat outside the scope of this particular manuscript.

Why the authors think that the Smc5/6 hinge can be more dynamic in loading/unloading based on the molecular latch in lieu of glycine motifs in Smc5? It appears that the latch might provide more stability. Please provide a clear rationale for this argument.

We have been very careful to indicate that this is a speculative comment. Our manuscript reads:

'We also speculate here that the distinctive structure of the North interface indicates (and facilitates) a more dynamic loading / unloading requirement for the Smc5/6 complex compared to cohesin or condensin'.

In support of our statement, we know there is a clear requirement for the Smc5/6 complex in S-phase; i.e. to stabilise stalled replication forks and/or to regulate any subsequent recombination steps if the fork has collapsed. These are intrinsically transient events – persisting for only relatively short periods of time – where we believe the Smc5/6 complex can rapidly engage / disengage from the replication fork. We therefore speculate that the specialised North interface (hinge AND hub) may facilitate this type of interaction.

Reviewer #3 (Remarks to the Author):

Multi-subunit Smc complexes fulfill diverse functions in genome maintenance. The Smc hinge domain is a central element of Smc, which dimerizes two Smc proteins to create Smc heterodimers (cohesin Smc1-3, condensin Smc2-4, and Smc5-6). This manuscript by the Oliver, Murray and Pearl labs elucidates for the first time the structure of the Smc5-6 hinge. As expected from low but significant levels of sequence similarity to other Smc hinge domains, the Smc5-6 hinge in the *S. pombe* crystal structure displays a typical Smc hinge fold. However, some clear deviations from the otherwise highly conserved architecture are obvious in Smc5-6. The authors describe these structural differences in detail. Using extensive mutagenesis, they assign functions to the newly identified structural elements: An exposed loop on Smc5 is shown to contribute to the dimerization interfaces. Consistently, it is found to be important for the survival in the presence of DNA damaging drugs. A loop on Smc6 is proposed to anchor the coiled coil onto the Smc5-6 hinge, while several arginine residues located in a patch of positively charged surface residues are being implicated in the preferential binding of the Smc5-6 hinge to single-stranded DNA. Small angle X ray scattering supports the existence of the crystallographically observed closed conformation of the Smc5-6 hinge dimer (and is inconsistent with an open form). This finding extends a recently proposed concept for condensin to the Smc5-6 complex. However, the functional implications are somewhat unclear.

Altogether, the report will be highly valuable for the field of Smc and chromosome biology. The manuscript is extensive but easy to follow. The data is well presented and generally makes a very solid impression. The conclusions from the work are valid. The manuscript should thus be a good candidate for publication in Nature Communications.

Specific comments

1. The R609E/R615E double mutant of Smc5 shows a strong growth defect, drug sensitivity and a DNA binding phenotype. Since both residues are located close to the dimerization interface, it is important to exclude the possibility that the observed effects are not indirectly caused by a dimerization defect. Please include the mutant in the pull-down assay (shown in Fig. 4D) or perform SAXS measurements.

We have confirmed that the R609E/R615E mutations do not affect hetero-dimerisation or conformation in solution (see responses to reviewers 1 and 2 above).

2. In the light of obvious similarity between the hinge-coiled coil arrangement (Fig. 7) in the new Smc5-6 hinge structure and in published Smc2-4 hinge structures, it is surprising that the Smc6 coiled coil (not Smc5 - the presumed homolog of Smc4) contacts the hinge domain. Could the authors speculate on whether this is due to convergent evolution (of hinge-coiled coil contacts) or whether Smc5 and Smc6 may possibly have been wrongly assigned to the sub-families of SMC proteins?

This is an interesting question, for which we don't have a simple answer. It is entirely plausible that the hinge-domains of Smc5 and Smc6 may have evolved along somewhat different and divergent pathways to their respective head domains; as presumably the only evolutionary pressure on them is maintenance of ATP-binding (at the head) and DNA-binding / dimerisation (at the hinge).

The work of Cobbe and Heck (Mol Biol Evol 2004) clearly indicates the problem when trying to classify and compare SMC proteins, 'Contrary to previous reports (Melby et al. 1998), we observed that eukaryotic SMC trees reconstructed from individual conserved hinge or N-terminal and C-terminal globular domains often differed both from each other and from phylogenies based on the whole protein sequence'.

So at the global amino acid sequence level, proteins may indeed appear to be paralogues of each other. But as Cobbe and Heck indicate, it is worth noting most of these analyses use the entire sequence of the protein, and rarely sub-divide into a protein's component sub-domains, where things become somewhat more complex.

3. Please include quality control experiments for the anti-Smc5 antibody used in Fig. 3B (the expression levels of exogenous protein seems surprisingly low compared to endogenous Smc5 despite the strong phenotype seen in Fig. 3C). Is the Smc5 mutant dominant in human cells but not yeast? If so, why?

We have repeated the western blot; using a different acrylamide gel formulation and running time to optimise separation of eGFP-fused Smc5 from the endogenous protein. A Western blot against cell extracts treated with siRNA targeting Smc5, confirms the antibody's specificity. Levels of the eGFP-fused protein are not affected as they are siRNA resistant, and a scrambled siRNA control has no visible effect on either protein.

We agree that the levels of exogenous protein (eGFP-fused Smc5) appear low, especially when compared to the endogenous protein. Experiments are currently ongoing in the laboratory to investigate this phenomenon.

We do not know whether the Smc5 mutant is dominant in yeast, as RMCE (recombination-mediated cassette exchange) is carried out in a haploid strain. The mutant strain does however, grow much slower than the WT and base-strains.

4. It would be highly desirable to have different names (such as A and A' for loops and residues)

and different color schemes on Smc5 and Smc6 throughout the manuscript. Currently, it is difficult to attribute residues and features to Smc5 or Smc6 hinge domains in some figures.

*We have amended our figures to ensure that (where appropriate) residues from Smc5 are consistently labelled in **bold** type, and those from Smc6 in italic type. Amino acids mutated in our study are additionally underlined. We also now use (as much as possible) two consistent colour schemes in our figures, either Scheme 1: Smc5 in cyan/blue, Smc6 in orange/yellow, plus any additional highlighted items in magenta, or Scheme 2: subdomain I in blue, subdomain II in grey, linker in cyan, loop A in green, loop B in orange, loop C in magenta.*

We have amended figure 1B to follow Scheme 1, which also serves to address the visibility issue with the connecting linker regions raised by Reviewer 1. Figure 5E has been recoloured, but by necessity is a hybrid of both colour schemes.

5. The title to Figure 7 is misleading since only a single conformation has been detected for Smc5-6 so far. Please modify accordingly.

We have amended the figure title to simply read "Conformation of the Smc5/6-hinge".

Reviewers' Comments:

Reviewer #2 (Remarks to the Author):

The authors have added new data as well as modified the text and figures/Supplemental figures to clarify most of the issues that reviewers have raised. One remaining concern regards the newly added Supplemental Figure 3A that shows mutant protein levels. This experiment needs to be repeated – the two wild-type lanes do not show the same protein levels; the blot appears to suggest that several mutants have much increased protein levels than wild-type. As increasing Smc5/6 levels along can cause toxicity, as shown previously, it is important to clarify whether mutant protein levels are indeed unchanged compared with wild-type protein.

Reviewer #3 (Remarks to the Author):

The manuscript by Alt et al. has been improved significantly during revision. All comments from the reviewers have been considered and many suggestions incorporated into the revised manuscript. It is now ready for publication in my opinion. However, it is worthwhile to explicitly state in the manuscript that it is currently unclear whether the closed conformations of Smc5-6 and condensin are evolutionarily related/conserved (see previous comment 2).

Reviewer #2 (Remarks to the Author):

The authors have added new data as well as modified the text and figures/Supplemental figures to clarify most of the issues that reviewers have raised. One remaining concern regards the newly added Supplemental Figure 3A that shows mutant protein levels. **This experiment needs to be repeated – the two wild-type lanes do not show the same protein levels; the blot appears to suggest that several mutants have much increased protein levels than wild-type. As increasing Smc5/6 levels along can cause toxicity, as shown previously, it is important to clarify whether mutant protein levels are indeed unchanged compared with wild-type protein.**

Expression levels of each mutant should be compared with the correct controls, in this case two base-strains used for recombination-mediated cassette exchange (smc5 WT | lox or smc6 WT | lox, with lox sites flanking the smc5 and smc6 loci containing the WT genes).

We have been careful to compare, by spot test, each base-strain back to the WT:JMM1188 control [see Fig3A, Figure5B, Figure5F, Supp. Fig3B, Supp. Fig5A] to look for any effects potentially caused by alterations in protein expression level. We see no differences / changes in sensitivity to any of the agents tested, and that any minor changes in expression level between these two control strains and WT produces no discernible phenotype. It is worth remembering that in this system all protein expression is driven from the native promoter, at the endogenous locus.

In Supplementary Figure 3A (Left), it is clear that all mutants are expressed to a similar level as the Smc5 WT | lox control, with the exception of HQD249: R587E/R619E which is apparently expressed at a lower level (yet has no apparent phenotype in our assays).

We erroneously included strain HQD89 in the western blot (lane 3, Supp. Fig 3A) and we believe that this is what may have caused confusion. This strain is not used in the experiments presented in this manuscript, and we have therefore removed it from both the supplementary figure and from the strain table. This matter aside, this particular strain still appears to express less protein than either JM1188:WT or HQD87:Smc6 WT | lox.

Similarly, in Supplementary Figure 3A (Right) – all mutants are expressed at similar levels to the Smc6 WT | lox control; except the temperature sensitive

*G551R strain; where **lower** level of the protein may correlate with destabilisation rather than expression.*

*We therefore in this instance, do not agree with the reviewer that any of the mutants have **much increased** protein levels. As a result, we do not believe the experiment requires or warrants repeating.*

Reviewer #3 (Remarks to the Author):

The manuscript by Alt et al. has been improved significantly during revision. All comments from the reviewers have been considered and many suggestions incorporated into the revised manuscript. It is now ready for publication in my opinion. **However, it is worthwhile to explicitly state in the manuscript that it is currently unclear whether the closed conformations of Smc5-6 and condensin are evolutionarily related/conserved** (see previous comment 2).

We thank the reviewer for their positive response, indicating that our manuscript should now be accepted for publication.

However, we do not fully understand the necessity or requirement for adding such a statement to our manuscript. We have been careful to compare the two SMC-complexes at just the structural / fold level – and to simply describe / highlight the similarities and differences between them. We do not make a distinct evolutionary argument at any point. We therefore politely decline this request.

Our previous response to the reviewer:

In the light of obvious similarity between the hinge-coiled coil arrangement (Fig. 7) in the new Smc5-6 hinge structure and in published Smc2-4 hinge structures, it is surprising that the Smc6 coiled coil (not Smc5 - the presumed homolog of Smc4) contacts the hinge domain. Could the authors speculate on whether this is due to convergent evolution (of hinge-coiled coil contacts) or whether Smc5 and Smc6 may possibly have been wrongly assigned to the sub-families of SMC proteins?

This is an interesting question, for which we don't have a simple answer. It is entirely plausible that the hinge-domains of Smc5 and Smc6 may have evolved along somewhat different and divergent pathways to their respective head domains; as presumably the only evolutionary pressure on them is maintenance of ATP-binding (at the head) and DNA-binding / dimerisation (at the hinge).

The work of Cobbe and Heck (Mol Biol Evol 2004) clearly indicates the problem when trying to classify and compare SMC proteins, 'Contrary to previous reports (Melby et al. 1998), we observed that eukaryotic SMC trees reconstructed from individual conserved hinge or N-terminal and C-terminal globular domains often differed both from each other and from phylogenies based on the whole protein sequence'.